# Elevated microglial oxidative phosphorylation and phagocytosis stimulate post-stroke brain remodeling and cognitive function recovery in mice

Shanshan Song [1,2✉], Lauren Yu[1,2], Md Nabiul Hasan[1,2], Satya S. Paruchuri[1,2], Steven J. Mullett[3,4], Mara L. G. Sullivan [5], Victoria M. Fiesler[1,2], Cullen B. Young[1,2], Donna B. Stolz[5], Stacy G. Wendell[3,4] & Dandan Sun [1,2,6✉]

New research shows that disease-associated microglia in neurodegenerative brains present features of elevated phagocytosis, lysosomal functions, and lipid metabolism, which benefit brain repair. The underlying mechanisms remain poorly understood. Intracellular pH ($pH_i$) is important for regulating aerobic glycolysis in microglia, where Na/H exchanger (NHE1) is a key pH regulator by extruding $H^+$ in exchange of $Na^+$ influx. We report here that post-stroke *Cx3cr1-Cre^{ER+/−};Nhe1^{flox/flox}* (*Nhe1* cKO) brains displayed stimulation of microglial transcriptomes of rate-limiting enzyme genes for glycolysis, tricarboxylic acid cycle, and oxidative phosphorylation. The other upregulated genes included genes for phagocytosis and LXR/RXR pathway activation as well as the disease-associated microglia hallmark genes (*Apoe, Trem2, Spp1*). The cKO microglia exhibited increased oxidative phosphorylation capacity, and higher phagocytic activity, which likely played a role in enhanced synaptic stripping and remodeling, oligodendrogenesis, and remyelination. This study reveals that genetic blockade of microglial NHE1 stimulated oxidative phosphorylation immunometabolism, and boosted phagocytosis function which is associated with tissue remodeling and post-stroke cognitive function recovery.

[1] Dept. of Neurology, Univ. of Pittsburgh, Pittsburgh, PA, USA. [2] Pittsburgh Institute for Neurodegenerative Disorders, Univ. of Pittsburgh, Pittsburgh, PA, USA. [3] Health Sciences Metabolomics and Lipidomics Core, Univ. of Pittsburgh, Pittsburgh, PA, USA. [4] Dept. of Pharmacology and Chemical Biology, Univ. of Pittsburgh, Pittsburgh, PA, USA. [5] Dept. of Cell Biology, University of Pittsburgh, Pittsburgh, PA, USA. [6] Veterans Affairs Pittsburgh Health Care System, Geriatric Research, Educational and Clinical Center, Pittsburgh, PA, USA. ✉email: songs2@upmc.edu; sund@upmc.edu

Metabolic reprogramming is essential for immune cells to regulate their effector responses[1,2]. The balance between glycolysis and oxidative phosphorylation is important for the transition between the damaging immune responses and their restorative functions in many immune cells[1,2]. Therefore, immune dysfunction/dysregulation is closely associated with loss of balance in immune energy metabolism homeostasis in many diseases, such as cancer, Alzheimer's disease, and stroke[2–4], and targeted regulation of immunometabolism emerges as a new therapeutic strategy for various types of neurodegenerative diseases[1,3]. Microglial cells, as the resident macrophages in the central nervous system, require a high energy expenditure to support their core functions such as surveillance and phagocytosis, where ATP is in high demand through various energy metabolism processes[5]. It is generally believed that homeostatic microglia primarily rely upon oxidative phosphorylation for ATP production, while inflammatory microglia reprogram their metabolism to suppress oxidative phosphorylation but stimulate aerobic glycolysis in both mouse models and human microglial cells[6,7]. Moreover, deficiency in microglial mitochondrial mass has been linked to microglial dysfunction and neurodegeneration in mice[4], revealing the significance of microglial energy metabolism fitness in maintaining their core functions, such as phagocytosis[4]. Microglial phagocytosis is directly involved in efficient clearance of myelin debris in support of white matter remyelination after demyelination injury[8,9], while dysfunction of microglial phagocytosis or inadequate elimination of the debris prolonged demyelination and/or impairs remyelination in either stroke or multiple sclerosis mouse models[10–12]. In addition, microglial phagocytosis impairment also limited sculpting of neural synapses/networks in synaptic pruning during early development or normal adolescent[13], or stripping the weak/injured synapses after ischemic stroke, contributing to cognitive decline in mouse models[14,15]. However, the underlying mechanisms of how microglia regulate their energy metabolism to support phagocytic functions remain poorly understood.

We previously reported that $Na^+/H^+$ exchanger isoform-1 (NHE1), which mediates $H^+$ efflux in exchange of $Na^+$ influx, is essential in regulating microglial homeostatic intracellular pH ($pH_i$) in primary microglial culture[16]. NHE1-mediated $H^+$ extrusion activity alkalinizes microglial $pH_i$ and promotes NADPH oxidase (NOX) function upon lipopolysaccharides (LPS) stimulation[16], or in mouse brains upon NMDA injection or after ischemic stroke[17]. Our recent study of selective deletion of microglial *Nhe1* in the *Cx3cr1-Cre$^{ER+/-}$;Nhe1$^{flox/flox}$* (cKO) mice demonstrated that loss of microglial NHE1 protein reduced pro-inflammatory microglial activation in ischemic brains and improved post-stroke motor-sensory functions[18]. In the current study, to investigate the underlying mechanisms, we conducted transcriptomic analysis of post-stroke wild-type (WT) and *Nhe1* cKO microglia by bulk RNA sequencing and measured microglial phagocytic activity, and cellular energy metabolism. We report here that the *Nhe1* cKO microglia displayed a stimulated spectrum of energy metabolism, featured with elevated transcriptomes for key rate-limiting enzymes involved in oxidative phosphorylation, as well as the tricarboxylic acid (TCA) cycle and glycolysis. Moreover, these cKO microglia showed a panel of upregulated genes for phagocytosis and liver X receptor-retinoid X receptor (LXR/RXR) pathway activation in the Ingenuity Pathways Analysis, which is also involved in microglial phagocytosis responses[19]. These changes were corroborated with the detection of increased oxidative phosphorylation capacity, elevated microglial phagocytic activity of bioparticles, enhanced synaptic remodeling in Golgi-Cox staining, and improved white matter myelination with transmission electron microscopy (TEM) and immunostaining. These findings suggest that microglial oxidative phosphorylation metabolism and elevated phagocytosis functions likely collectively enhance brain tissue remodeling and post-stroke cognitive function recovery. We identified the pH-regulatory protein NHE1 as a modulator for microglial immunometabolism and function in brain tissue repair.

## Results

### Microglial *Nhe1* cKO mice displayed accelerated post-stroke cognitive function recovery.

We previously reported that *Cx3cr1-Cre$^{ER+/-}$;Nhe1$^{f/f}$* mouse line successfully deleted NHE1 protein expression only in IBA1$^+$ microglia, but remained intact in other cell types[18]. WT and *Nhe1* cKO mice underwent ischemic stroke injury (by the well-established transient middle cerebral artery occlusion model, tMCAO[18]). Mortality, changes of body weight and neurological behavior functions were monitored during day 1-28 post-stroke period (Fig. 1a, Supplementary Figure 1). Sham procedures did not induce any mortality in either WT or cKO mice, while stroke led to ~18% mortality in WT but 0% in cKO mice ($p = 0.13$, Supplementary Figure 1a). Both WT and cKO mice exhibited ~20% body weight loss after stroke injury (Supplementary Figure 1b-c). However, the cKO mice displayed faster post-stroke body weight recovery especially in the female mice ($p = 0.038$, Supplementary Figure 1d). Both sexes showed similar results otherwise, and thus were pooled for analysis. These data indicate that the *Nhe1* cKO mice are more tolerant to ischemic stroke. This is corroborated by our recent observation that *Nhe1* cKO mice showed better sensorimotor function in the acute to subacute post-stroke phase (1-14 d)[18]. In this study, we further assessed whether cognitive functions of WT and *Nhe1* cKO mice are differently affected by stroke in the y-maze spontaneous alternation test, the novel object recognition test, and the open field test (Fig. 1a). Figure 1b–c shows that sham-operated WT and cKO mice had comparable levels of triad spontaneous alternation rates, while the cKO stroke mice displayed a significantly increased rate of spontaneous alternation compared to the WT stroke mice, suggesting a stimulated short-term spatial working memory[20] in the cKO mice. In addition, these cKO mice also showed significantly improved long-term recognition memory in the novel object recognition test (Fig. 1d–e). Compared to sham controls, stroke reduced the discrimination index (from 0.19 to 0.06), and recognition index (from 0.60 to 0.53) in WT mice. However, the cKO mice showed significant increases in both discrimination index and recognition index compared to the WT mice ($p = 0.023$), indicating improved abilities of the cKO mice in recognizing and distinguishing new objects from the old ones. Moreover, the WT and cKO mice exhibited similar locomotor activity after sham procedures (Supplementary Fig. 1e, Fig. 1f–g), while the post-stroke WT mice displayed hyperactivity reflected by the increased travel distance compared to the post-stroke cKO mice in the open field test (Fig. 1f–g). Taken together, these data demonstrate that the *Nhe1* cKO mice exhibited better post-stroke recovery evidenced with faster post-stroke body weight recovery, improved long-term and short-term memory function recovery, and ameliorated hyperactivity.

### Microglial *Nhe1* cKO mice displayed increased microglial transcriptomes for phagocytosis and cholesterol export.

To better understand the underlying mechanisms of improved poststroke cognitive function recovery in the *Nhe1* cKO mice, we performed microglial transcriptomic analysis by bulk RNAseq of P2RY12$^+$ microglia isolated by magnetically-activated cell sorting (MACS) from non-lesion (contralateral, CL) and ischemic (ipsilateral, IL) hemispheres of WT and cKO brains at 3-day post-stroke (Fig. 2a). The purity and specificity of these microglia were confirmed by high expression of several microglia-specific markers, along with low levels of cell type markers for neurons, astrocytes, or oligodendrocytes (Supplementary Figure 2a-c). It

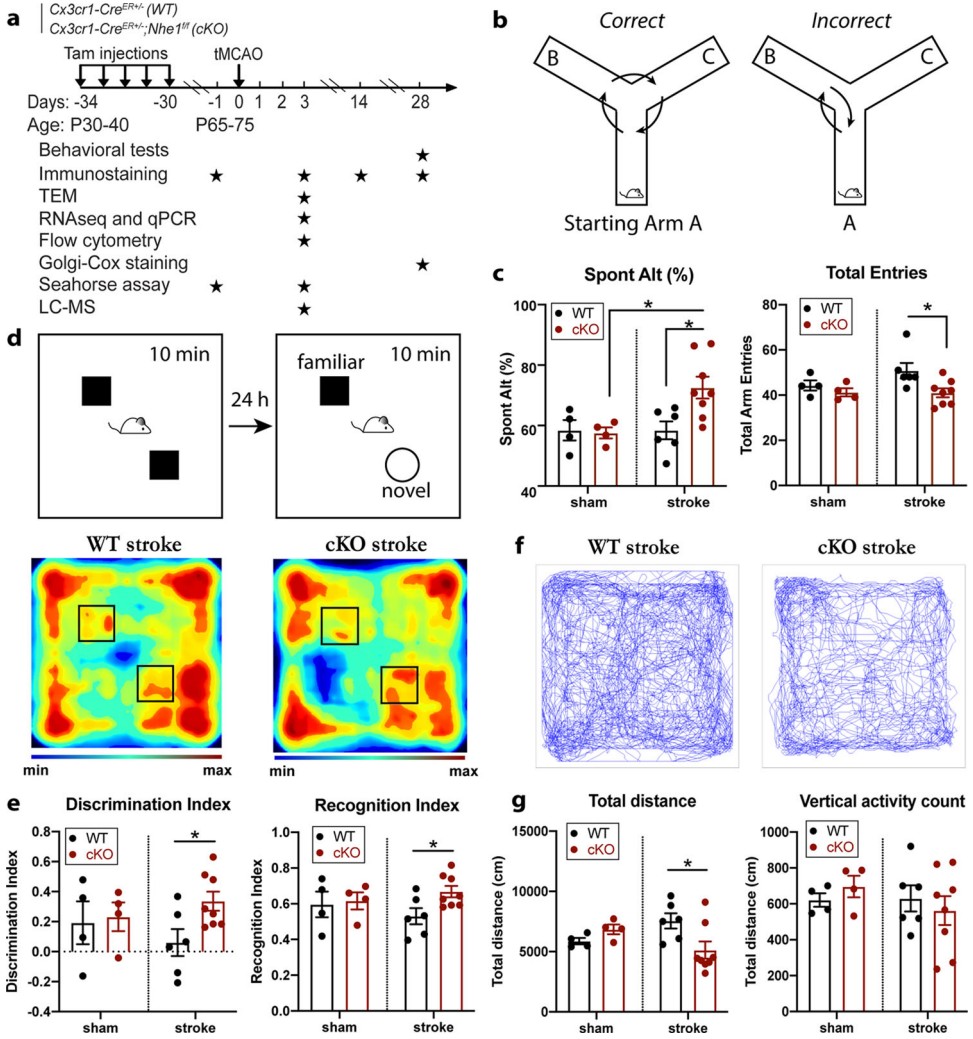

**Fig. 1 Microglial *Nhe1* cKO mice displayed improved post-stroke cognitive functions. a** Experimental timeline. **b** Illustration of Y-maze spontaneous alternation test. **c** Percentages of triad spontaneous alternation and total arm entries in WT and cKO mice at 28 d post-sham or stroke. **d** Illustration of novel object recognition (NOR) test and representative heatmaps of mice exploring the objects. **e** Calculated discrimination index and recognition index. **f** Representative paths of WT and cKO mice in the open field (OF) test over a 60-min duration. **g** Total traveled distance and counts of vertical activity. N = 4 animals for sham groups, and N = 6–8 animals for stroke groups. Data are mean ± SEM. *p < 0.05.

was reported recently[21] that P2RY12 protein remained abundant and detectable in proinflammatory, phagocytosing microglia in brains of Alzheimer's disease, where P2RY12 microglial cells were positive for CD68, progranulin, and HLA-DR[21]. Consistent with these findings, we detected similar level of *P2ry12* gene expression in WT or cKO microglia from our bulk RNAseq, and comparable CD11b⁺CD45⁺P2RY12⁺ microglial population in flow cytometry between the WT and cKO brains post-stroke (CL and IL hemispheres, Supplementary Figure 2d). These findings collectively indicate that our isolated P2RY12⁺ microglial population from WT and cKO stroke brains represent the microglia population in both control and stroke contexts. Principle component analysis showed that microglial mRNA transcriptome profiles from CL or IL hemispheres of WT and cKO brains were separated by 60.9% of their top principal components (Fig. 2b). Heatmaps and volcano plots illustrated the patterns of up- and downregulated genes (with fold change > 2 and p value < 0.05) in the CL and IL hemispheres of cKO brains, compared to the WT brains (Fig. 2c–d). Figure 2e further depicted the stroke-induced up- or downregulated genes in the WT and cKO brains (IL vs. CL), respectively. By comparing differentially expressed genes (DEGs) in each and every comparison of WT and cKO microglia,

we identified 128 common DEGs (up- or downregulated) in microglia from both CL and IL hemispheres of the cKO brains (Fig. 2f, **yellow dashed area**), while 219 DEGs (up- or downregulated) were exclusively expressed in the CL hemispheric microglia of the cKO brains (Fig. 2f, **dashed red area**), and 370 DEGs (up- or downregulated) in the IL hemispheric microglia of the cKO brains (Fig. 2f, **dashed green area**). Analyzing the 370 DEGs for downstream pathways using IPA, we found that a majority of the pathways (15/28) was involved in inflammation and/or immune responses. Interestingly, the top upregulated pathways included significant LXR/RXR activation (p = 7.15E-03), whose functions in microglia include regulating the expression of phagocytosis genes such as MerTK, and mediating transrepression at pro-inflammatory promoters[19], along with the phagosome maturation pathways (Fig. 3a). Critical genes within the LXR/RXR pathway activation were significantly upregulated in cKO microglia compared to WT (p = 0.0014, Fig. 3b), including *Nr1h3* (encoding LXRa), *Apoe*, *Abca1*, *Abcg1*, and *Mertk*, where they work together to induce more phagocytosis of debris, including stroke-induced myelin debris which is rich in cholesterol[22], and cholesterol export activity (Fig. 3c). Moreover, genes involved in phagocyte recruitment, recognition, and

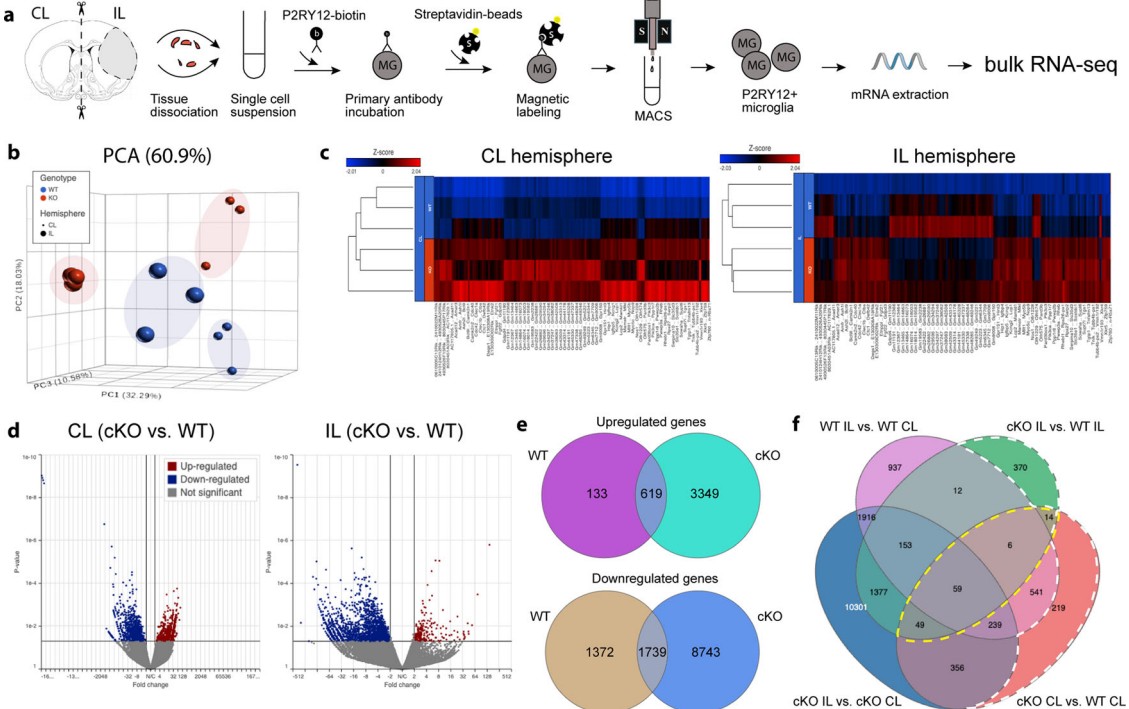

**Fig. 2 Post-stroke differential transcriptomic changes of P2RY12+ microglia from WT and *Nhe1* cKO mice. a** Schematic illustration of P2RY12+ microglia isolation for bulk RNAseq on Illumina HiSeqX platform. **b** Principle component analysis plot showed separation of WT (blue) or cKO (red) data by their top three principal components. CL: small dots. IL: large dots. *N* = 3 biologically independent animals. **c** Heatmaps and unsupervised hierarchical clustering showing hemispheric microglial gene cluster changes. **d** Volcano plots of up- and downregulated genes. **e** Venn diagrams showing stroke-induced up- and downregulated genes in WT or cKO microglia (IL vs. CL). **f** Venn diagram with identified genes of interests for downstream pathway analysis.

engulfment (Fig. 3d), as well as in phagolysosomal function and lipid metabolism (related to the digestion process of phagocytosis) were also significantly enhanced in the cKO microglia from both CL (Supplementary Fig. 3) and IL hemispheres (*p* < 0.0001, Fig. 3e–f).

A recent report identified a new population of microglia in the context of Alzheimer's disease, termed disease-associated microglia, with distinct features of elevated phagocytosis, phagolysosomal function, and lipid/cholesterol metabolism[23], which shared many molecular signature markers with microglia in neurodegenerative diseases (MGnD)[24]. However, in our MCAO-induced stroke model, these gene signatures were elevated in both the lesioned IL hemisphere, and the non-lesion CL hemisphere. While microglial responses are largely dependent on specific disease contexts, virtually all of the disease-associated microglia/MGnD signature genes reported[23,24] were significantly elevated in the cKO stroke microglia (*p* = 0.0001, Fig. 3g). Using qRT-PCR in independent sets of RNA samples, we successfully validated the mRNA expressions for multiple important genes for phagocytosis, LXR/RXR pathway activation, and disease-associated microglia/MGnD signatures (Fig. 3h). Lastly, we conducted targeted LC-MS/MS for detection of free cholesterol content within isolated P2RY12+ microglia from WT or cKO brains, and observed that the cKO microglia exhibited similar levels of total cholesterol compared to the WT microglia (Fig. 3i).

**Microglial *Nhe1* cKO brains exhibited increased post-stroke microglial phagocytic activity and improved dendritic spine plasticity.** We further tested whether the elevated transcriptome for all phagocytosis processes at 3-day post-stroke in Fig. 3 would translate into increased phagocytic activity in these cKO microglia. We previously reported similar infarct volume between the WT and cKO post-stroke mice[18], therefore we assessed microglial

phagocytic activity in hemispheric brain tissues in WT and cKO brains. Figure 4a-b illustrated that the *E.coli*-FITC+ bioparticle uptake occurred in the CD11b+/CD45+/P2RY12+ microglial population in WT brains (82.7 ± 6.7%). In contrast, nearly all of the CD11b+/CD45+/P2RY12+ microglial population (97.3 ± 0.6%) in post-stroke cKO brains displayed phagocytic activity of the bioparticles (*p* = 0.0047). In addition, microglial phagocytic activity in the non-stroke CL hemisphere from cKO brains was also significantly increased compared to the WT brains (Fig. 4b). These ex vivo data suggest that selective deletion of microglial *Nhe1* enhances their phagocytic activity in the non-lesion, as well as in the acute stroke tissues, which is in line with our RNAseq transcriptomic findings. However, additional in vivo assessment of microglial phagocytic activity is warranted in the future study.

As microglial phagocytosis is important for synapse pruning/remodeling in both healthy immature and mature brains[25–27], or in injured brains[14,15], we next investigated the impact of increased microglia phagocytosis on engulfment and clearance of the injured synaptic structures at the acute post-stroke phase and on the chronic post-stroke synapse remodeling. C1q is the initiating protein of the classical complement cascade, where its cleaved products attract phagocytes and tags target cells for elimination by phagocytosis, thus has been used as a marker for targets of microglial phagocytosis[28]. Of note, C1q is increased and associated with synapses in the context of Alzheimer's disease[29]. Figure 4c shows that C1q associated with the post-synaptic density (PSD95) (C1q+/PSD95+ colocalizing puncta) were detected in the peri-lesion cortex of WT brains (~24% of total puncta) at 3 d post-stroke. Interestingly, the cKO mice exhibited increased number of C1q+/PSD95+ colocalizing puncta by 2.2-fold (*p* = 0.005) (arrows, Fig. 4c–e), along with phagocytosing microglial morphology showing increased C1q expression (arrowhead, Fig. 4d), an indication of microglia-mediated

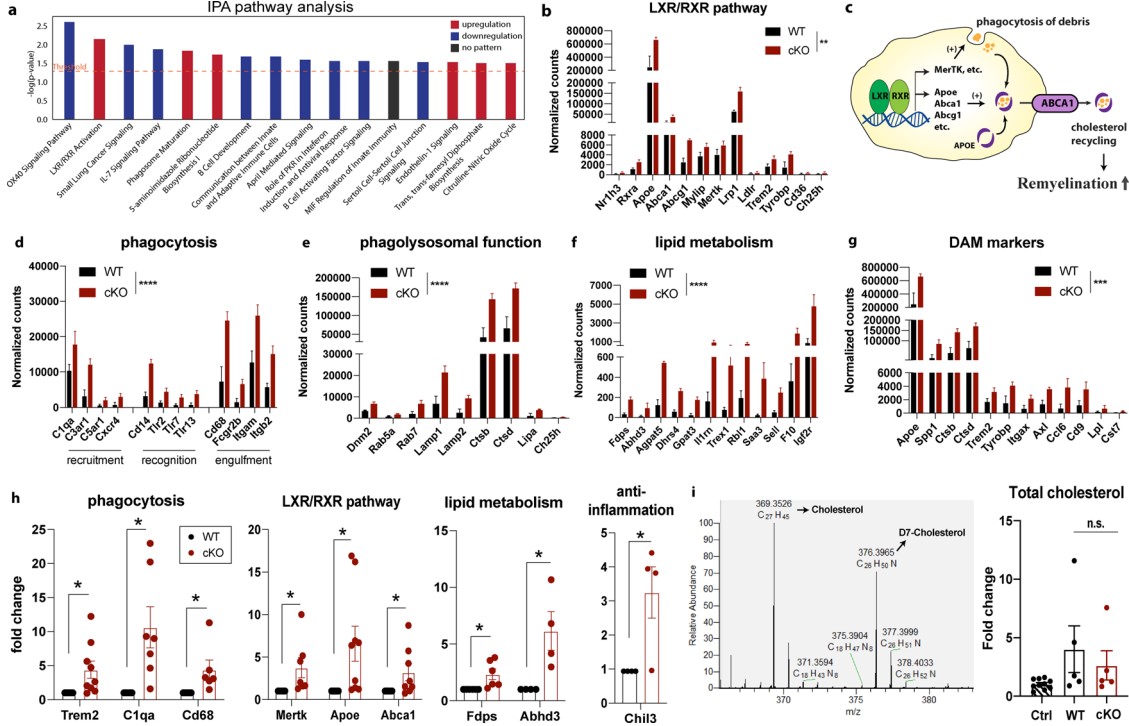

**Fig. 3 Upregulated transcriptomes of phagocytosis and LXR/RXR activation in the *Nhe1* cKO microglia. a** Differentially up- and downregulated pathways from Ingenuity Pathway Analysis revealed elevated phagocytosis and LXR/RXR pathway activation in the cKO microglia. **b** Transcriptome analysis of genes regulated by LXR/RXR pathway in the IL hemispheres of WT and cKO microglia. **c** Illustration of the LXR/RXR activation in the regulation of phagocytosis and cholesterol metabolism. **d–g** Transcriptome analysis of genes relating to phagocytosis processes (recruitment, recognition, and engulfment), phagolysosomal function, lipid metabolism, and disease-associated microglia markers. $N = 3$ animals. **h** qPCR validation of genes for phagocytosis processes (*Trem2, C1qa, Cd68, Mertk*), LXR/RXR pathway activation (*Apoe, Abca1, Mertk*), lipid metabolism (*Fdps, Abhd3*), and restorative microglial marker (*Chil3*, for Ym1 protein) in the P2RY12+ microglia. $N = 4$-9 animals. **i** Measurement of total cholesterol content (normalized to internal control D7-cholesterol) in isolated P2RY12+ microglia from WT and cKO brains at 3-day post-stroke by LC-MS. Nonstroke CL hemisphere was used as control (Ctrl). $N = 5$ biologically independent samples. Data are mean ± SEM. *$p < 0.05$, **$p < 0.01$, ***$p < 0.001$, ****$p < 0.0001$.

phagocytosis of synapses/dendritic spines[29]. Supplementary Figure 4 in assessing changes of the lysosomal marker LAMP1 expression further revealed that the inclusion of PSD95+ synapses within the LAMP1+ cells was more frequently detected in cKO post-stroke brains, compared to the WT brains ($p < 0.0001$). These data strongly suggest that deletion of microglial *Nhe1* enhanced microglial phagocytosis for stripping of injured dendritic spines during the acute phase post-stroke. More importantly, these changes may impact synapse remodeling in the chronic post-stroke recovery stage (Fig. 4f–m). Golgi-Cox staining and Imaris analysis of the peri-lesion cortical layer V pyramidal neurons revealed more damaged/broken neuronal dendrites of WT brains at 28-day post-stroke (**arrows**, Fig. 4f–g), while minimal damaged morphology was observed in the cKO brains (**arrowhead**, Fig. 4f–g). The layer V pyramidal neurons of the cKO brains exhibited increased number of total branch points ($p < 0.0001$, Fig. 4h) as well as increased total dendritic length ($p < 0.0001$, Fig. 4i). Sholl analysis of these cortical neurons highlighted significant increases in dendritic branching from the proximal (15 μm) to distal (145 μm) areas to the neuronal body in the cKO brains ($p < 0.05$, Fig. 4j). Importantly, a dramatic reduction in spine density was observed in the WT mice compared to the cKO mice at 28-day post-stroke ($p < 0.0001$; Fig. 4k–l). Moreover, morphology analysis of these dendritic spines revealed a significant increase in the proportion of mushroom-shaped spines ($p = 0.0052$), with reduced thin/filipodia spines ($p = 0.015$) in the cKO brains, compared to the WT (Fig. 4m). The stubby type of spines remains unchanged between the two groups ($p = 0.34$, Fig. 4m). These findings

provide additional evidence that elevated microglial phagocytosis in the cKO brains at the early phase post-stroke could facilitate synaptic stripping for better synaptic remodeling.

Considering communications between microglia and infiltrated peripheral monocytes[30], we further assessed changes of macrophages, neutrophil, and T cells in the spleen and their brain infiltration in WT and cKO mice with flow cytometry (Supplementary Figure 5a). Supplementary Figure 5b showed that similar cell counts of spleen CD11b+CD45hi macrophages, CD11b+CD45hiLy6G+ neutrophils, and CD3+ T cells were detected in WT and cKO sham mice. At 3-day post-stroke, no significant changes of spleen immune cells were detected in WT or cKO mice. These findings are consistent with reports that stroke did not significantly change the number of macrophages and neutrophils in the spleen[31]. In contrast, Supplementary Figure 5c showed that stroke led to an increased brain infiltration of macrophages, neutrophils, and T cells in WT as well as cKO mice, similar to other reports of stroke brains[31,32]. Taken together, our data imply that selective deletion of *Nhe1* in microglial cells did not significantly change either immune cell infiltration in stroke brains or immune responses in the spleen.

***Nhe1* cKO microglia exhibited boosted oxidative phosphorylation metabolism transcriptome and function**. How *Nhe1* cKO microglia increased their phagocytic function remains unknown. Within the 128 common DEGs from both hemispheres, our Gene Ontology analysis and Gene Set Enrichment Analysis showed significantly altered metabolism spectrum between WT and *Nhe1* cKO microglia (Fig. 5a-b). We found

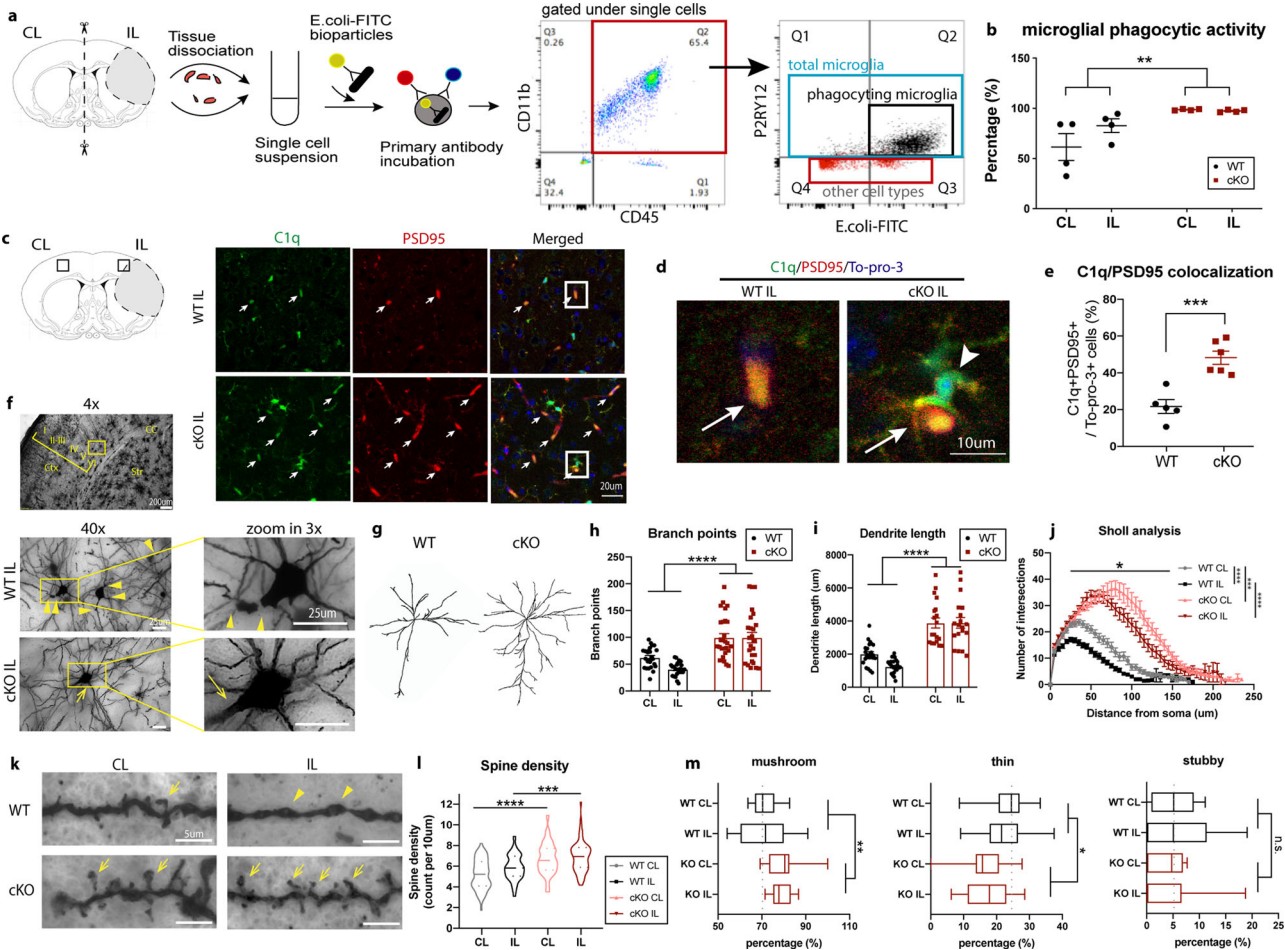

**Fig. 4 *Nhe1* cKO brains exhibited increased post-stroke microglial phagocytic activity and improved dendritic spine plasticity. a** Isolation of CD11b⁺/CD45⁺/P2RY12⁺ microglial cells and gating strategies with engulfed *E.coli*-FITC⁺ bioparticles from the CL and IL hemispheres of WT and cKO brains at 3-day post-tMCAO. P2RY12⁺ containing *E.coli*-FITC⁺ population: Q2 (black). Nonmicroglial cells: Q3-4 (red) with a population of YFP⁺ cells in the FITC channel. **b** Quantification of microglial phagocytic activity. N = 4 animals. **c** Representative staining images of C1q, PSD95, and To-pro-3 from the IL peri-lesion area at 3-day poststroke. Arrows: C1q⁺/PSD95⁺ colocalizing cells. **d** Enlarged images of colocalized C1q and PSD95 in IL peri-lesion brain area (from boxed areas in C). Arrow: C1q⁺/PSD95⁺ colocalizing cells. Arrowhead: C1q⁺ microglial cell. **e** Quantitative analysis of C1q⁺/PSD95⁺ colocalizing cells. N = 5–6 animals, 3 images per animal. **f** Representative 40x Golgi-Cox staining images of layer V neurons in the IL peri-lesion cortex of WT and cKO brains at 28-day post-stroke. Arrowheads: damaged/broken dendritic branches. Arrows: preserved/regrown dendritic branches. **g** Representative neuron morphology in the IL peri-lesion cortex of WT and cKO brains at 28-day post-stroke. **h** Sholl analysis of branching intersections in relation to distance from soma in neurons in the CL and IL peri-lesion cortex of WT and cKO brains at 28-day post-stroke. N = 24–28 neurons, from 4 brains per group, 3 areas per brain, 2–3 images per area. **i–j** Dendritic branch points and dendritic length from the same cohort of neurons as in h. **k** Representative 100x images of dendritic spines from the layer V neurons in the CL and IL peri-lesion cortex of WT and cKO brains at 28-day post-stroke. **l** Spine density from the secondary or tertiary dendrites were analyzed in same cohort of neurons as in h. Images were taken with z-stacked projection to include all spines from the dendrite. **m** Characterization of spine morphology. 100x images were analyzed as in k. Mushroom, stubby, and thin spines were counted and expressed as the percentage of all spine counts. Data are mean ± SEM. *p < 0.05, **p < 0.01, ***p < 0.001, ****p < 0.0001.

upregulated energy metabolism in multiple processes in *Nhe1* cKO microglia, including glycolysis (p < 0.0001), the TCA cycle (p < 0.0001), and oxidative phosphorylation (p < 0.0001) (Fig. 5c–e). Importantly, multiple genes encoding mitochondrial complexes I–V were significantly upregulated (p < 0.0001, Fig. 5e). We next performed metabolism assays using Seahorse Extracellular Flux Analyzer, with pooled microglial cells from the CL hemispheres of WT and cKO stroke mice as the control groups, as no differences in microglial glycolytic metabolism were detected in naïve WT and cKO mice (Supplementary Figure 6). Seahorse mitochondrial stress tests showed that ischemic stroke stimulated basal and maximal mitochondrial respiration rates in WT microglia (p = 0.0063 and p = 0.018, respectively; Fig. 5f–g), possibly due to feedback mechanisms to ramp up microglial activation under inflammatory environment[33]. Interestingly,

compared to WT microglia, the cKO microglia further accelerated basal and maximal mitochondrial respiration, as well as ATP-linked respiration (p = 0.0003, p = 0.0007, and p = 0.0057, respectively; Fig. 5f–g). Moreover, Fig. 5h–j showed that microglial cells from the nonlesion CL hemispheres displayed low basal glycolysis-mediated extracellular acidification rate (ECAR), an indirect analysis of the glycolytic rate of cells[34]. Stroke triggered significant stimulation of glycolysis in WT microglia, reflected by the increased basal glycolysis, as well as glycolytic capacity (Fig. 5h–i). In contrast, cKO microglial cells exhibited significantly slower ECAR, indicating less H⁺ extrusion rate and reduced glycolysis (Fig. 5h–i). Overall, the bioenergetics profile of WT and cKO microglia in Fig. 5j clearly demonstrated that microglia from the non-lesion CL brain tissues are less energetic with low respiration and glycolysis. WT microglia of stroke brains

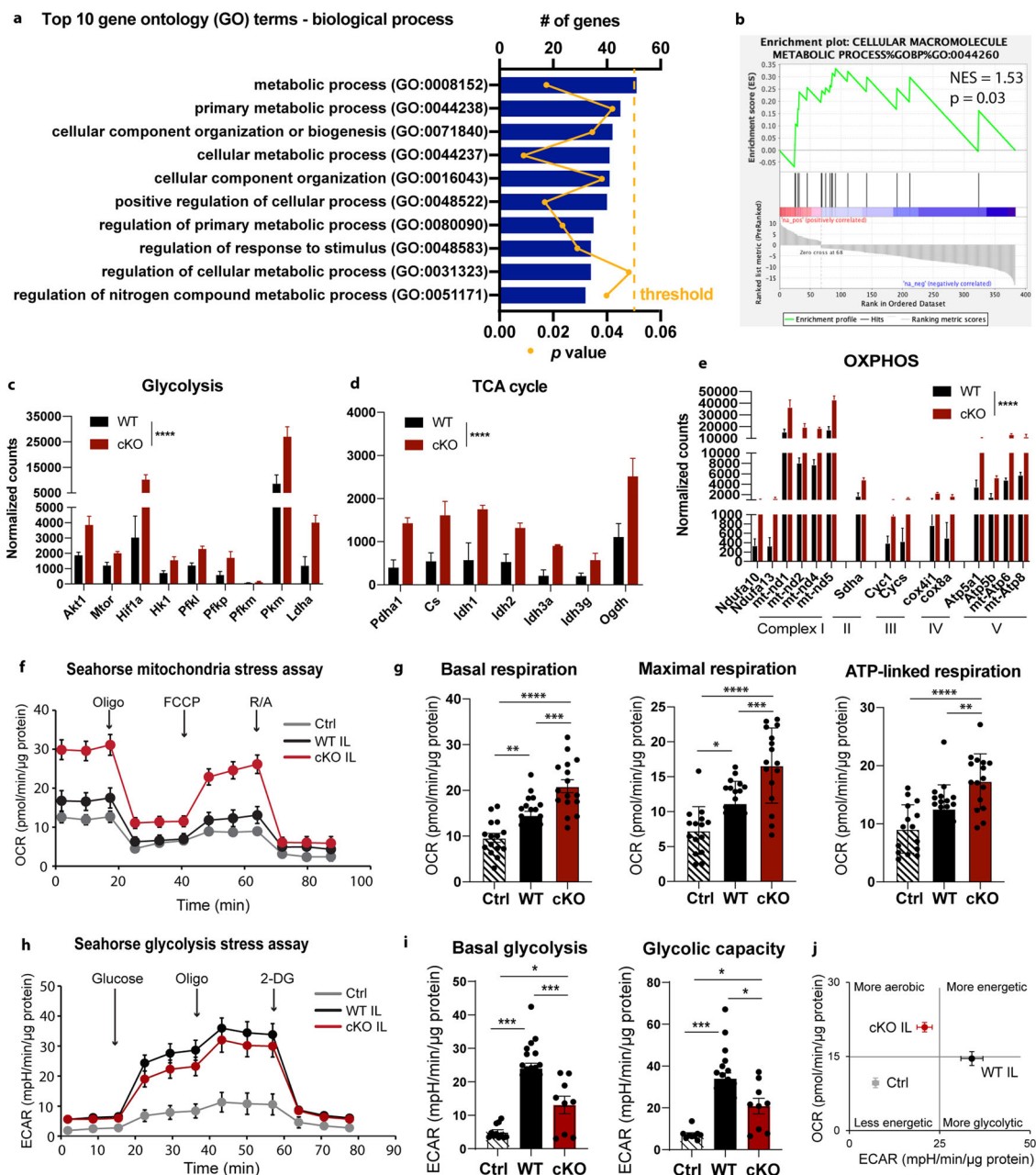

**Fig. 5 *Nhe1* cKO microglia showed accelerated oxidative phosphorylation and ATP-linked respiration. a–b** Top 10 Gene Oncology terms in biological process, and Gene Set Enrichment Analysis analyzed with DEGs from Fig. 2D. **c–e** Transcriptome analysis of genes relating to glucose metabolism processes, including glycolysis, TCA cycle, and oxidative phosphorylation. *N* = 3 animals. **f–g** Oxygen consumption rate (OCR) measured by mitochondrial stress test using Seahorse Extracellular Flux Analyzer in isolated P2RY12+ microglial cells. As we observed similar results in non-lesion CL hemispheres from WT and cKO brains, these samples were pooled to represent as the non-lesion control (Ctrl). *N* = 15-20 measurements, from 3-4 independent experiments. **h–i** Extracellular acidification rate (ECAR) measured by Seahorse Extracellular Flux Analyzer in isolated P2RY12+ microglial cells. CL hemispheres from WT and cKO were pooled as the non-lesion control (Ctrl). *N* = 9-20 measurements, from 3 independent experiments. **j** Snapshot of bioenergetic profile of P2RY12+ microglia cells. Data are mean ± SEM. **$p < 0.01$, ***$p < 0.001$, ****$p < 0.0001$.

significantly ramped up their glycolytic activity than mitochondrial oxidative phosphorylation and thus showed more glycolytic phenotype. In contrast, the cKO microglia of stroke brains relied more on the oxidative phosphorylation for their energy demand (Fig. 5j). These data suggest that reduced glycolysis in the cKO microglia after stroke likely resulted from less NHE1-mediated H+ extrusion and more acidic pH$_i$, preventing Warburg effect on stimulating glycolysis[35]. These data strongly suggest that loss of microglial NHE1 protein in the *Nhe1* cKO brains plays an important role in fine-tuning microglial glycolytic and oxidative

phosphorylation metabolism to provide ATP fuels in meeting their energy demands in phagocytic functions.

***Nhe1* cKO mice showed stimulated white matter myelination and oligodendrogenesis.** We previously detected high number of mature oligodendrocytes (APC+) in the corpus callosum (CC) and external capsule (EC) of cKO brains at 14 d post-stroke, which correlated with poststroke functional improvement[18]. However, the underlying mechanisms are not well understood. Accumulating evidence shows that microglial phagocytosis plays

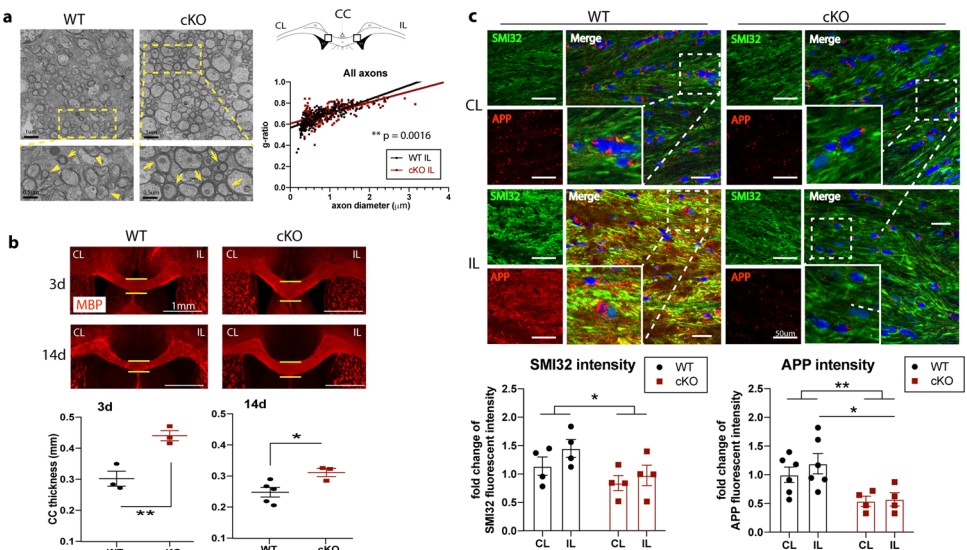

**Fig. 6 Microglial *Nhe1* cKO mice exhibited increased white matter resistance to ischemic stroke. a** Representative images and g-ratio from axons in the medial corpus callosum (CC) by transmission electron microscopy (TEM). $N = 30$ myelinated axons per image, 3 images per animal, from 3 biological independent animals per group; totaling ~270 myelinated axons per group. Slopes from the linear regressions in WT or cKO mice were compared. **b** Increased CC thickness (at same bregma level) in *Nhe1* cKO mice at 3 and 14 d post-stroke by MBP staining. $N = 3–5$ animals. **c** Representative images and quantitative analysis of SMI32 and APP intensity in CL and IL peri-lesion area of CC from WT or *Nhe1* cKO mice at 3 d post-stroke. $N = 4–6$ animals. Data are mean ± SEM. *$p < 0.05$, **$p < 0.01$.

an important role in the clearance of myelin debris and stimulating remyelination[8,9]. We thus tested whether cKO mice exhibited stimulated oligodendrogenesis and white matter myelination. Indeed, we detected that in all myelinated axons from the medial portion of CC, myelin integrity was severely disrupted and the slope of g-ratio relative to axon diameter was significantly increased for the stroke-injured hemisphere of WT mice, indicating thinner myelin for a given axon diameter[36] in the WT mice, but not in the post-stroke cKO mice (Fig. 6a) or in the non-stroke hemispheres of either WT or cKO mice (Supplementary Fig. 7a). Especially, the difference was most apparent in small to medium-sized axons while in large axons there was no difference (Supplementary Fig. 7b). As reported, in the CC of normal adult mice, approximately 60–70% of axons are unmyelinated and these axons do not exceed 0.6 µm in diameter[37]. Thus, the greater myelination of small to medium diameter axons could result from myelination of these axons that were not myelinated in normal circumstances[37], which was also correlated with behavioral improvement[37]. We also detected significantly enhanced myelination of CC in the cKO brains (midline thickness at the same bregma level (Supplementary Fig. 7c) at 3–14-day post-stroke compared to WT brains (Fig. 6b). Moreover, as another integral part of white matter tissues, axons can also present damages in demyelination lesions, which may represent a stronger correlation with functional loss in human MRI studies[38]. We detected significantly decreased accumulation of SMI32 and APP intensities in the CC of cKO brains compared to the WT brains ($p = 0.031$ and $p = 0.0025$, respectively; Fig. 6c), both of which are indications of damaged axons and have been found in demyelinating lesions[39].

In assessment of oligodendrogenesis and cell death, quantification of OPC counts (NG2$^+$Olig2$^+$), proliferative OLs (Ki67$^+$Olig2$^+$), and apoptotic OLs (Caspase3$^+$Olig2$^+$) were conducted at baseline (naïve, Supplementary Fig. 7d), and 3–28-day post-stroke in the CC of WT and cKO brains (Fig. 7a–c). Figure 7a–c **(first column)** and Supplementary Fig. 7d showed higher percentage of OPCs (NG2$^+$Olig2$^+$; 1.5-fold, $p = 0.016$) and lower apoptotic OL populations (Caspase3$^+$Olig2$^+$; 0.5-

fold, $p = 0.0031$) in the CC of naïve cKO brains than the WT brains, while similar counts of proliferative OLs (Ki67$^+$Olig2$^+$; $p = 0.25$) were detected in the naïve WT and cKO brains. These naïve cKO brains concurrently exhibited increased thickness of CC (Supplementary Figure 7d). On the other hand, stroke led to a 51.8 % initial decrease of NG2$^+$Olig2$^+$ OPCs in WT brains at 3-day post-stroke, which did not further decline at 14–28-day post-stroke (Fig. 7a). In comparison, the cKO brains showed significantly higher levels of NG2$^+$Olig2$^+$, Ki67$^+$Olig2$^+$, than the WT brains throughout 3–28-day post-stroke (Fig. 7a–b), with elevated NG2$^+$Olig2$^+$ and Ki67$^+$Olig2$^+$ counts both at the acute 3-day post-stroke (2.3-fold, $p = 0.0005$ and 2.8-fold, $p < 0.0001$, respectively) and the 28-day post-stroke chronic phase (1.8-fold, $p = 0.036$ and 1.3-fold, $p = 0.016$, respectively; Fig. 7a–b). The apoptotic Caspase3$^+$/Olig2$^+$ cells in the WT brains occurred early at 3-day post-stroke (2.8-fold, $p < 0.0001$) and remained significantly elevated at 28-day post-stroke (2.5-fold, $p = 0.0069$) (Fig. 7c). Interestingly, the cKO brains exhibited transient but delayed apoptosis of the Olig2$^+$ cells at 14 d post-stroke, which were quickly reduced to the baseline level by 28-day poststroke (Fig. 7c). Because we detected an increased number of APC$^+$ mature oligodendrocytes in the cKO brains at 14 d post-stroke[18], we further tested whether the differentiation rate was also accelerated in these oligodendrocyte lineage cells from an earlier phase in cKO brains. Indeed, H3 lysine 9 trimethylation (H3K9me3), a post-translational histone modification marker for OPC differentiation[40], was concurrently elevated in these Olig2$^+$ cells in the cKO brains at 3-day post-stroke (1.4-fold, $p = 0.030$), further evidence of accelerated OPC differentiation in the cKO brains (Fig. 7d). The ECs of WT and cKO brains show similar changes in NG2$^+$Olig2$^+$, Ki67$^+$Olig2$^+$, and Caspase3$^+$Olig2$^+$ cell counts as in CC (Supplementary Fig. 8). Taken together, our study clearly demonstrates that specific deletion of microglia *Nhe1* stimulates early oligodendrogenesis and OPC differentiation under normal physiological and ischemic conditions that result in increased resistance/tolerance of white matter to ischemic injury.

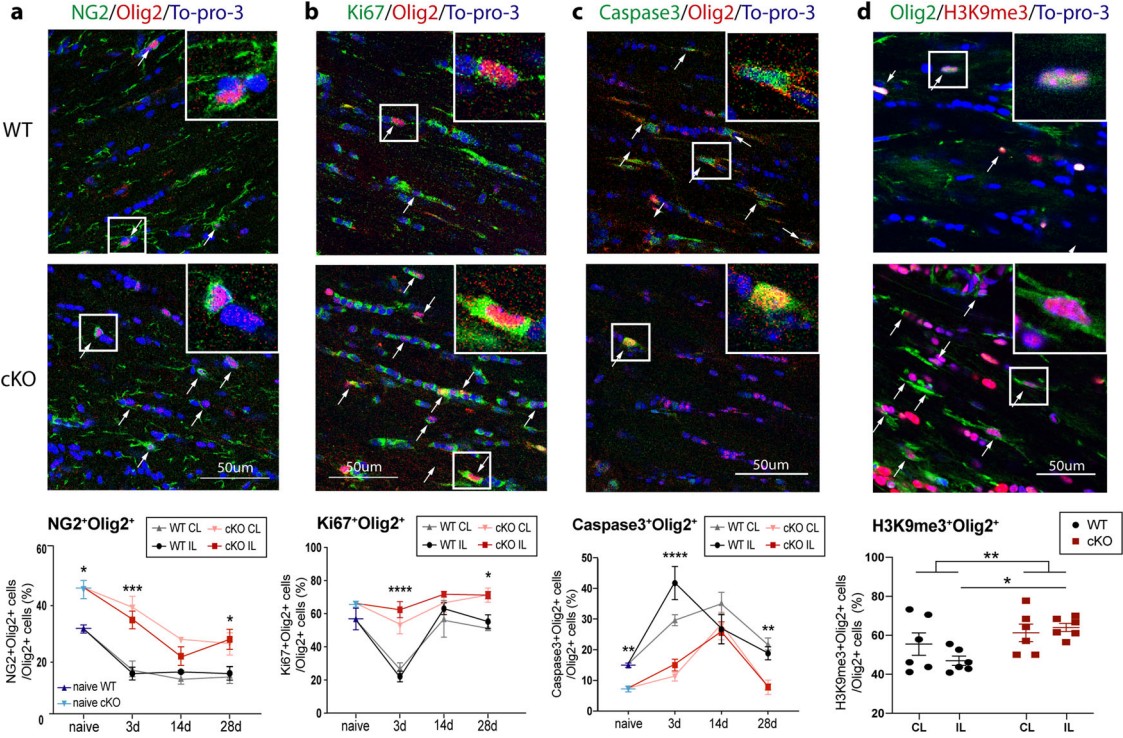

**Fig. 7 Increased oligodendrogenesis contributed to enhanced myelination in the microglial *Nhe1* cKO mice. a–c** Representative images and quantitative analysis of NG2$^+$/Olig2$^+$, Ki67$^+$/Olig2$^+$, and Caspase3$^+$/Olig2$^+$ cells in CL and IL hemispheres of CC. **d** Changes of OPC differentiation marker H3K9me3 expression in CC at 3 d post-stroke. $N = 3$ animals for naïve (non-stroke) mice; $N = 6$–7 animals for 3d post-stroke; $N = 3$–4 for 14d post-stroke; $N = 3$–4 for 28d post-stroke. Data are mean ± SEM. $*p < 0.05$, $**p < 0.01$, $***p < 0.001$, $****p < 0.0001$.

## Discussion

We observed that the post-stroke cKO microglia exhibited increased transcriptome profiles for all processes in phagocytosis (recruitment, recognition, engulfment, and digestion), and stimulated LXR/RXR pathway activation, a regulatory pathway that stimulates phagocytosis and promote anti-inflammatory responses in microglial cells[19]. As a result, we concurrently detected increased phagocytic activity in these cKO microglia, and enhanced oligodendrogenesis and remyelination in poststroke cKO mice. It is worth mentioning that circulating monocytes can also become phagocytic and beneficial for debris clearance upon ischemic injury[30]. However, we did not detect differences in CD11b$^+$CD45$^{high}$ macrophage cell counts or other infiltrated immune cells between the WT and cKO brains through 3–14-day post-stroke (Supplementary Fig. 5). These findings suggest that the microglial NHE1 protein is involved in modulating microglial phagocytic function, which subsequently impacts post-injury brain repair.

Accumulating evidence suggests that disease-associated microglia are activated by sensing the neurodegeneration-associated molecular patterns in the brain, such as apoptotic neurons, myelin debris, and lipid degradation products[41], and play a protective role by expressing a unique signature of genes relating to phagocytosis and lipid metabolism[23], such as *Apoe, Ctsd, Lpl, Tyrobp, Trem2, Cd9*, and *Cst7*, etc. Interestingly, the *Nhe1* cKO microglia showed similar upregulation of these genes after ischemic stroke. These unbiased data indicate that suppression of microglial *Nhe1* promotes microglial transcriptomes upon ischemic stroke injury which resemble those disease-associated microglia phenotypes. A more precise transcriptome profiling of these *Nhe1* cKO microglial cells with a better separation of different subcluster populations is needed, which can be investigated with single-cell RNAseq in future studies.

Additionally, the LXR/RXR pathway is also known for its key roles in regulating cholesterol efflux activity, especially in astrocytes and

neurons[19]. While it remains unclear how LXR/RXR activation may impact cholesterol transport in microglial cells, an increased accumulation of cholesterol in microglial cells could induce formation of proinflammatory microglial foam cells and/or prohibit cholesterol recycling to neurons and oligodendrocytes for remyelination[42,43]. A significantly elevated anti-inflammatory marker *Chil3* (encoding Ym1 protein) was detected in the post-stroke cKO microglia, indicative of a transformation to the restoring microglial phenotype[44]. Moreover, compared to its non-lesion hemisphere, or to the post-stroke WT microglia (Fig. 2d–e), the post-stroke cKO microglia displayed a large number of downregulated genes that were mostly involved in inflammatory responses, such as the OX40 signaling pathway (Fig. 3a) involving in NF-kB activation[45]. This is consistent with the previous report that NHE1 blockade inhibited NF-kB activation[46]. However, we did not detect significant changes of the total cholesterol content in WT and cKO microglia; this could be masked by a dynamic balance of enhanced phagocytosis (of cholesterol-rich myelin debris) and concurrently elevated cholesterol export activity, both of which are downstream functions of the LXR/RXR activation[19]. Additional measurements on the specific changes in cholesterol metabolism as well as transport activities in WT and cKO microglia are warranted in future studies.

We also compared stroke-induced differential pathway changes between CL and IL hemispheres from WT or cKO brains using IPA (Supplementary Fig. 9a). While stroke mainly induced upregulation of the coagulation system pathway in WT brains (Supplementary Fig. 9b), as expected in the ischemic stroke condition[47], the cKO microglia also upregulated several signaling pathways after stroke, including the eIF2 signaling pathway and oxidative phosphorylation pathway (Supplementary Fig. 10c). Inactivation of eIF2 was associated with neurodegenerative diseases such as Alzheimer's disease[48], and was linked to leukoencephalopathy featured with brain white matter lesion[49]. The mechanisms underlying the upregulation of these transcriptions

in cKO microglia and its impact on tissue repair should be investigated in future studies.

Healthy microglial phagocytosis function is required for selectively engulfing weak or injured pre- and post-synaptic elements during development[26,50]. After brain injury, microglial phagocytic activity is essential for sculpting neural synapses/networks and increasing synchronized neuronal firing by stripping axosomatic synapses[14,15,51]. We detected increased C1q/PSD95 co-localization in the *Nhe1* cKO brains at 3 d post-stroke, where C1q, the initiating protein in the classical complement cascade, can selectively target synaptosomes expressing proteins involved in apoptotic processes that need to be eliminated by microglia, and thus has emerged as a critical mediator for synaptic refinement and plasticity[25,27,28]. In addition to stimulated microglial stripping of the damaged dendritic spines in the acute post-stroke *Nhe1* cKO brains, we also detected enhanced spine remodeling at the chronic stage of 28-day post-stroke, reflected by the increased number of dendritic branching, as well as a higher dendritic spine density and restored spine morphology, which was positively correlated with the improved cognitive memory functions in these cKO mice (Supplementary Figure 10). Regarding the better performance of post-stroke cKO mice than post-sham group in the y-maze test, the possible causes are not clear. The previous report has shown that an elevation in BDNF levels in the post-stroke aged animals may underlie the neural circuitry enhancement that contributes to their improvement in cognitive flexibility, compared to the aged sham controls[52]. Our cKO stroke brains indeed showed improved synaptic remodeling for network strengthening. Future studies are warranted to examine whether changes of BDNF levels contribute to the enhancement of neural circuit in these cKO mice. On the other hand, the locomotor activity in the open field test showed the post-stroke WT animals were hyperactive while the post-stroke cKO mice were not. This could be due to the significantly increased mushroom-shaped dendritic spines in the cortex of cKO brains, which has been described as the mature "memory spines" to benefit cognitive memory functions[53]. The restoration of microglial engulfment function is reported to rescue aberrant spine morphology and improve cognitive functions[54], while deficiency of microglia-mediated elimination of synaptic structures failed in synapse refinement, and led to changes in brain functional connectivity and resulted in diseased/abnormal behavior[55]. Our findings strongly suggest that selective deletion of microglial *Nhe1* promoted stripping of dendritic spines and synapses through increased microglial phagocytosis, which could contribute to faster post-stroke cognitive function recovery.

White matter lesions are the most common pathology feature of vascular contributions to cognitive impairment and dementia[56]. Persistent demyelination in neurodegenerative diseases can be caused by excess accumulation of myelin debris with inefficient clearance or dysfunction of microglial phagocytic activity, while the resulting axonal dystrophy, inhibited OPC differentiation, and mature OL reduction collectively impaired the remyelination process[12,57]. We observed that the cKO mice exhibited increased white matter repair (both in myelination and axonal integrity) and synapse remodeling, along with significantly improved cognitive function. We suspect that this was largely resulted from early microglia-mediated phagocytosis of myelin debris in the cKO brains. Additional studies are needed to determine the temporal course of the microglial phagocytic activity and correlation to white matter repair.

Lastly, we speculate that the impact of selective deletion of *Nhe1* in cKO microglial functions results from NHE1-mediated regulation of $pH_i$. We reported that NHE1 activity is essential to regulate basal microglial $pH_i$ (at $\sim7.19 \pm 0.03$[16]). NHE1-mediated $H^+$ extrusion activity was stimulated and shifted the microglial $pH_i$ to $7.29 \pm 0.02$ ($p < 0.05$) in response to LPS[16], and promoted NOX function[16,17] and aerobic glycolysis in proinflammatory microglial activation[16,58]. In general, intact mitochondrial TCA cycle and oxidative phosphorylation are required for microglia/macrophage differentiation[59] and other microglial functions, including phagocytosis[3]. Suppression of oxidative phosphorylation leads to diminished microglial immune responses, such as cytokine secretion and phagocytosis[60]. Interestingly, our study shows that compared to WT controls, the *Nhe1* cKO microglia displayed 2-4 folds increase in all the key rate-limiting enzyme genes for TCA cycle (PHDA1, CS, IDH, OGDH) and oxidative phosphorylation complex I-V as well as for glycolysis (HK1, PFK1, PKM2). These cKO microglia concurrently displayed higher oxidative phosphorylation capacity and ATP-linked respiration, as well as reduced basal glycolysis and glycolytic capacities. These findings strongly suggest that deletion of *Nhe1* acidifies microglia, which prevents excessive $pH_i$ alkalization and glycolysis but utilizes oxidative phosphorylation to fuel the restorative microglial phagocytosis activity. Thus, our study identified NHE1 protein as a modulator for the dynamic tuning of microglial immunometabolism and function in brain tissue repair. Of note, $pH_i$ homeostasis is involved in regulating energy metabolism as well as transcriptome changes[61]. More acidic $pH_i$ has been linked to stimulating the master transcription regulator *Hif1a*[61]. Whether these biochemical mechanisms play a role in the transcriptome changes in cKO microglia is warranted for further study. In addition, pharmacological approaches using potent NHE1 inhibitors such as Cariporide or Rimeporide[62–64] should be employed to assess their efficacy in stimulating microglial energy metabolism and phagocytosis in post-stroke brain repair.

In summary, we discovered that transgenic deletion of the major microglial pH-regulating protein NHE1 switched microglial metabolism from glycolysis to oxidative phosphorylation to generate more ATP. These microglia concurrently showed elevated phagocytosis activity, and enhanced synaptic remodeling and myelin repair. Improved post-stroke cognitive function recovery was observed following these metabolic alterations (Fig. 8). These findings identify NHE1 protein as a modulator for dynamic tuning of the microglial immunometabolism and functions in brain tissue repair, with therapeutic potentials for neurological diseases with dysregulated microglial functions.

## Methods

**Animals.** All animal studies were approved by the University of Pittsburgh Medical Center Institutional Animal Care and Use Committee, which adhere to the National Institutes of Health Guide for the Care and Use of Laboratory Animals, and reported in accordance with the Animal Research: Reporting In Vivo Experiments (ARRIVE) guidelines[65]. A total of 212 mice (male and female, 2–3 months old) were used in the study. Animals were provided with food and water ad libitum and maintained in a temperature-controlled environment in a 12/12 h light-dark cycle. All efforts were made to minimize animal suffering and the number of animals used.

*Cx3cr1-CreER*$^{+/-}$ (wild-type, WT) control mice and *Cx3cr1-CreER*$^{+/-}$;*Nhe1*$^{f/f}$ (*Nhe1* cKO) mice were established as described previously[18]. Both genotypes of mice (male or female) at postnatal day 30–40 (P30-40) received tamoxifen (Tam, Sigma) (75 mg/kg body weight/day at a concentration of 20 mg/ml in corn oil, intraperitoneally) for 5 consecutive days. As *Cx3cr1* is expressed by both brain resident microglia and peripheral infiltrating bone marrow-derived myeloid cells (BMDM), a 30-day postinjection waiting period was given for clearance of Tam[66–68] and for replenishing of *Cx3cr1*$^+$ monocytes[69] prior to induction of ischemic stroke (Fig. 1a), considering the *Cx3cr1*$^+$ BMDM, but not the *Cx3cr1*$^+$ microglia, have a 21-day self-renewing cycle[70]. This method has been proven to achieve a complete replenishment of *Cx3cr1*$^+$ BMDM[69] and is effective in our study[18,71] as well as in others[72–74]. Surgeries and all outcome assessments were performed by investigators blinded to mouse genotype and experimental group assignments.

**Transient focal ischemic stroke model.** Focal cerebral ischemia was induced by occlusion of the left middle cerebral artery (MCA) as described before[18,75]. Briefly,

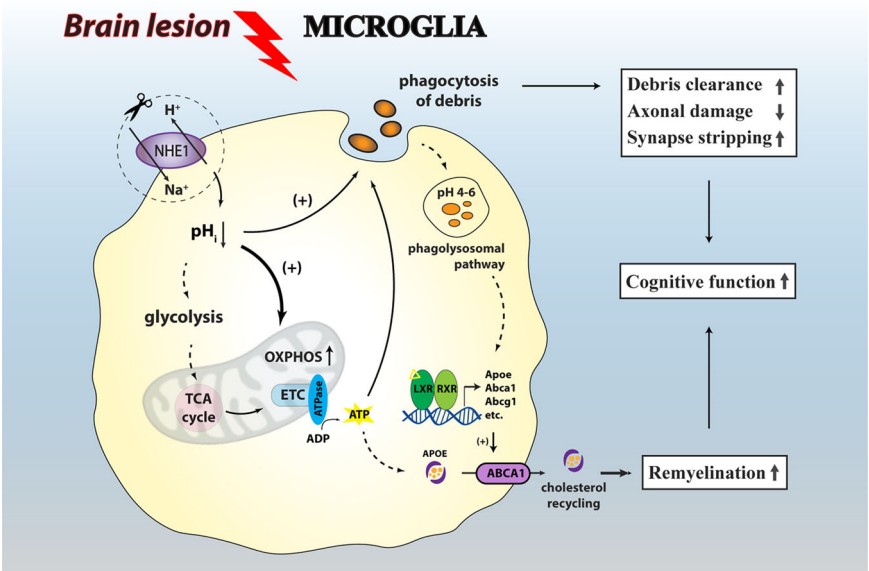

**Fig. 8 Illustration of NHE-1-mediated microglial pH$_i$ regulation and immunometabolism in ischemic brain repair.** Brain injury stimulates NHE1 protein expression in microglia, which mediates H$^+$ efflux in exchange of Na$^+$ influx and alkalinizes pH$_i$ in microglia. Selective deletion of microglial *Nhe1* acidifies pH$_i$, which is involved in boosting the immunometabolism of oxidative phosphorylation, as well as ATP production, which provides fuels for important microglial functions such as phagocytosis. Increased phagocytosis of debris improved debris clearance in favor of remyelination, reduced axonal damage, and promoted synaptic stripping for remodeling. The acidified pH$_i$ could also potentially benefit the phagolysosomal pathway to provide ligand for the activation of LXR/RXR pathway, which increase transcription of cholesterol transporter genes and promote cholesterol recycling to support remyelination. These changes may collectively contribute to cognitive function improvement after brain lesions.

mice were kept under 1.5% isoflurane anesthesia during the procedure and the core temperature (37.0 °C) was maintained by a small animal temperature controller pad throughout all procedures. After midline skin incision, the left common carotid artery was exposed and the superior thyroid artery and occipital artery branches of the external carotid artery were isolated and coagulated. The animals were subjected to MCA occlusion (MCAO) by the introduction of a silicone-coated suture (6-0 monofilament nylon, Doccol, USA) inserted via the external carotid artery. Reperfusion was established by the withdrawal of the filament after 60 min of transient MCAO (tMCAO). The incision was closed and the mice were allowed to recover under a heating lamp to maintain the core temperature (36~37 °C) during a 30-60 min recovery period. Sham controls underwent identical procedures without the use of suture. All animals were returned to their cages with free access to food and water after the procedures.

**Neurological function tests**. Neurological functional deficits in mice were screened in a blinded manner. Cognitive functions were measured with the Y-maze spontaneous alternation test and the novel object recognition test, and locomotor functions were measured with the open field test, all previously reported for identifying and quantifying neurological functional deficits in ischemic white matter injury rodent models[76].

a. *Y-maze test*. The Y-maze spontaneous alternation test was used to assess spatial working memory, as described previously[77]. Each animal was placed into one arm of the Y-maze and monitored over an 8-min duration with an overhead video tracking system at 25 d post-stroke. A sequential list of arms entries was analyzed using an automated Sequence Analysis Tool macro in excel. Spontaneous alternation was only counted when a mouse entered three different arms consecutively. The spontaneous alternation % was calculated as the percentage of the number of triad spontaneous alternation in total arm entries minus two.

b. *Open field test*. The open-field test was used to detect gross locomotor activity, as described previously[78]. Each mouse was placed in the center of an open field chamber (50 cm × 50 cm × 50 cm) and monitored for 60 min with an overhead video tracking system at 26 d post-stroke. Each animal's total travelled distance and vertical activity were recorded. The open-field test also served as the habituation session for novel object recognition test, as described below.

c. *Novel object recognition test*. The novel object recognition test was used to examine different memory processes, including acquisition, consolidation, and retrieval, as described previously[79]. Open field test performed at 26-day poststroke were used as habituation session to the empty arena. For training, each mouse was placed in the center of the same arena at 27 day poststroke, and allowed to explore two identical objects placed at opposite quadrants of the arena. Each animal will be monitored for 10 min with the overhead

video tracking system. For testing of long-term recognition memory, at 24 h after the training session (28-day poststroke), one of the training objects was replaced with a novel object, and the mouse was monitored for another 10 min. Each animal's path and time spent around each object were recorded. Discrimination index was calculated as $(T_{novel} - T_{old})/(T_{novel} + T_{old}) \times 100\%$, and recognition index was calculated as $T_{novel}/(T_{novel} + T_{old}) \times 100\%$. Mice with a total exploration time for both objects less than 20 s in either training or testing sessions were excluded from the study, as it could not be confirmed that they spent enough time to learn or discriminate, as described before[79].

**Flow cytometry**. Mice were euthanized with overdose of CO$_2$ and transcardially perfused with ice-cold saline, as described before[18]. After removal of cerebellum and meninges, CL and IL hemispheric tissues were separated and dissociated into single-cell suspensions using a neural tissue dissociation kit with the gentleMAC Octo Dissociator (Miltenyi Biotech Inc., Germany). Myelin was removed using the 30/70 Percoll gradient method as described[18]. Microglial phagocytic activity was determined by incubating cells (10$^6$ cells/mL) with FITC-conjugated *E.coli* bioparticles (1 mg/mL) (Thermo Fisher Scientific, USA) for 1.5 h at 37 °C, as described[80,81], and subsequently stained with BV421-conjugated CD11b (1:400, BioLegend, USA), PerCP-Cy5.5-conjugated CD45 (1:400, BioLegend, USA), and APC-conjugated P2RY12 (1:400, BioLegend, USA) antibodies for 20 min at 4 °C in the dark. Microglial phagocytic activity was measured as the percentage of bioparticle$^+$/CD11b$^+$/CD45$^+$/P2RY12$^+$ phagocytosing microglia within CD11b$^+$/CD45$^+$/P2RY12$^+$ total microglia, using an LSRFortessa flow cytometer (BD Biosciences, USA) running FACS Diva software (BD Bioscieinces, USA) with the following settings: Forward scatter (FSC) V = 600, mode = Lin; Side scatter (SSC) V = 275, mode = Lin; FITC V = 450, mode = Log; BV421 V = 400, mode = Log; PerCP-Cy5.5 V = 500, mode = Log; APC V = 400, mode = Log. In each experiment, at least 10,000 events were recorded from each hemispheric sample for analysis. For the analysis of the splenic and brain infiltrated immune cells, flow cytometry of the spleen and brain samples was performed. Spleens were homogenized and filtered through a 70 µM strainer and red blood cells were lysed with ACK lysing buffer (Gibco, USA). Brain samples were dissociated into the single-cell suspension as described above. Cells were stained with FITC-conjugated CD45 (1:250, Invitrogen, USA,), BUV395-conjugated CD11b (1:250, BD Biosciences), APC-conjugated P2RY12 (1:250, Biolegend USA), PerCP-Cy5.5-conjugated Ly6G (1:250, Biolegend USA), and BV421-conjugated CD3 (1:250, Biolegend USA) antibodies and measured with the following settings: Forward scatter (FSC) V = 370, mode = Lin; Side scatter (SSC) V = 320, mode = Lin; FITC V = 500, mode = Log; BUV395 V = 480, mode = Log; APC V = 390, mode = Log; PerCp-Cyanine5.5 V = 550, mode = Log; BV421 V = 360, mode = Log. In each experiment, at least

20,000 events were recorded from each hemispheric sample for analysis, and data were analyzed using FlowJo (BD Biosciences, USA) software.

**Microglia isolation by magnetic-activated cell sorting (MACS).** Single-cell suspensions from the CL and IL hemispheric tissues were collected at 3 d post-tMCAO as described above, and P2RY12$^+$ microglia were isolated by MACS using a MojoSort mouse P2RY12 selection kit (BioLegend, USA). Briefly, the single-cell suspensions were incubated with biotin-conjugated P2RY12 for 20 min at 4ºC, then incubated with streptavidin-conjugated nanobeads for 20 min at 4ºC, before washing through an MS column (Miltenyi Biotech Inc., Germany) placed on a OctoMACS magnetic field separator (Miltenyi Biotech Inc., Germany), with the magnetically labeled P2RY12$^+$ cells maintained in the column and unlabeled P2RY12$^-$ cells washed away. The purity of isolated P2RY12$^+$ microglia was determined with subsequent labeling of BV421-conjugated CD11b, and expressed as the percentage of CD11b$^+$ population within the total P2RY12$^+$ population. The P2RY12$^-$ cells were also collected to monitor purity and yield. Samples with a purity percentage > 95% were subsequently used for RNA sequencing and bioinformatic analysis.

**Bulk RNA sequencing and bioinformatics analysis.** Bulk RNA sequencing and bioinformatic analysis were performed in P2RY12$^+$ microglia samples sorted by MACS and paired-end sequenced in the platform of Illumina HiSeq X, as in our recent report[82]. Briefly, the library was prepared using the SMARTer stranded total RNA pico V2 kit (Takara Bio, USA). Adapter sequences, low-quality reads, and contamination of ribosomal RNA and mitochondrial DNA were removed prior to alignment to the mouse genome (mm10) using the STAR aligner (2.5.3a) in Partek Flow 8.0 software (Partek, USA). Quantification was performed by annotation to the mouse genome and DESeq2 was performed for the DEG analysis. Genes with $p$ value < 0.05 and fold change > 2 or < −2 were considered as differentially expressed. Gene Ontology analysis of the DEGs was perform on Panther Classification System[83], and Gene Set Enrichment Analysis was conducted[84]. Ingenuity Pathway Analysis (Qiagen Bioinformatics, Germany) was conducted to identify enriched biological pathways. The identified transcriptome profiles were subsequently verified by real-time qPCR (described below). The sequencing data have been deposited to the Gene Expression Omnibus database with experiment series accession number GSE175504.

**Quantitative real-time qPCR.** The identified transcriptome profiles from the RNAseq bioinformatic analysis were subsequently verified by quantitative real-time qPCR. P2RY12$^+$ microglia were isolated as described above, and microglial RNA was isolated with an RNeasy Plus Micro Kit (Qiagen, Germany), following the manufacturer's instruction. The RNA concentration was quantitated using a NanoDrop 1000 spectrophotometer (Thermo Fisher Scientific, USA). Equal amount of RNA was processed for cDNA synthesis using the iScript cDNA synthesis kit (Bio-rad, USA). qPCR was performed using the iTaq Universal SYBR Green Supermix (Bio-rad, USA) on a CFX96 Real-Time PCR Detection System (Bio-rad, USA). Primer sequences are as follows: Apoe forward: CTCCCAAGTC ACACAAGAACTG, reverse: CCAGCTCCTTTTTGTAAGCCTTT; Abca1 forward: GCTTGTTGGCCTCAGTTAAGG, reverse: GTAGCTCAGGCGTACAGAG AT; Trem2 forward: CTGGAACCGTCACCATCACTC, reverse: CGAAACTCGA TGACTCCTCGG; Mertk forward: CAGGGCCTTTACCAGGGAGA, reverse: TG TGTGCTGGATGTGATCTTC; Cd68 forward: TGTCTGATCTTGCTAGGACCG, reverse: GAGAGTAACGGCCTTTTTGTGA; Cq1a forward: AAAGGCAATCCA GGCAATATCA, reverse: TGGTTCTGGTATGGACTCTCC; Chil3 forward: ACC TGCCCCGTTCAGTGCCAT, reverse: CCTTGGAATGTCTTTCTCCACAG; Cyp 8b1 forward: CAGAGAAAGCGCTGGACTTC, reverse: GGCCCCAGTAGGGAG TAGAC; Fdps forward: GGAGGTCCTAGAGTACAATGCC, reverse: AAGCCT GGAGCAGTTCTACAC; GAPDH forward: AACTTTGGCATTGTGGAAGG, reverse: ACACATTGGGGGTAGGAACA; Hnrnpab forward: ATGGCGGCTA CGACTACTC, reverse: GCTGGCTCTTTCCGTAATTTGT. Data were analyzed using the ΔΔCt method[85] with triplicate reactions for each gene evaluated and normalized to GAPDH or Hnrnpab as the internal control.

**Extracellular flux analysis of P2RY12$^+$ microglia cells.** Mitochondrial stress tests were performed using Seahorse extracellular flux analyzer (Agilent Technologies) per the manufacturer's instruction, as we reported recently[82]. P2RY12$^+$ microglial cells were isolated from 3-day poststroke brain tissues using MACS, as described above. $1 \times 10^5$ cells per well were seeded in an XF96 well plate in DMEM base media. The OCR was measured with the addition of (i) 2 μM Oligomycin, (ii) 1 μM Carbonyl cyanide-4-(trifluoromethoxy) phenylhydrazone, (iii) 1 μM Antimycin A + Rotenone. The ECAR was measured with the addition of (i) 10 mM glucose, (ii) 2 μM Oligomycin, (iii) 50 mM 2-Deoxyglucose. Measurements were done using an XF96 extracellular flux analyzer and results were analyzed using Wave v.2.2.0 software.

**Liquid chromatography–mass spectrometry (LC-MS).** Intracellular total cholesterol in P2RY12$^+$ microglia were measured by HPLC-MS/MS. P2RY12$^+$ microglia were isolated at 3 d post-stroke using MACS, before resuspended in 100 μL cold PBS and spiked with cholesterol-d$_7$ (1.27 nmoles, Avanti Polar Lipids,

Alabaster, AL) internal standard and Folch extracted with the addition of 900 μL water, 1 mL methanol, and 2 mL chloroform/formic acid. Samples were centrifuged at 3000 rpm at 4 ℃ for 10 min. The bottom layer (organic) was transferred to a clean vial and dried under N$_2$. Samples were reconstituted in 100 μL of chloroform for high-performance liquid chromatography-electrospray ionization tandem mass spectrometry analysis (LC-MS/MS).

Analyses were performed by untargeted LC-HRMS. Briefly, Samples were injected via a Thermo Vanquish UHPLC and separated over a reversed-phase Phenomenex Kinetex C8 column (2.1 × 100 mm, 3 μm particle size) maintained at 55 ℃. For the 30 min LC gradient flowing at 0.3 mL/min, the mobile phase consisted of the following: solvent A (10 mM ammonium formate /0.1% formic acid in 6:4 ACN/H$_2$O) and solvent B (10 mM ammonium formate /0.1% formic acid in 90:10 IPA:ACN). The gradient started at 5% B for 2 min and increased to 10% B over the next 4 min. This was followed by an increase to 15% B over 4 min, and an increase to 50% B over 3.5 min. The organic was increased to 95% B over 9.5 min. and held for 3 min before equilibration at 5% B for 4 min. The Thermo ID-X tribrid mass spectrometer was operated in positive ESI mode. A data-dependent MS2 method scanning in Full MS mode from 300 to 800 $m/z$ at 120,000 resolution with an AGC target of 5e4 for triggering MS2 fragmentation using stepped HCD collision energies at 25, 30, and 35 in the orbitrap at 15,000 resolution. Source ionization spray voltage setting was 3.5 kV. Source gas parameters were 35 sheath gas, 5 auxiliary gas at 300 ℃, and 1 sweep gas. Calibration was performed prior to analysis using the Pierce™ FlexMix Ion Calibration Solutions (Thermo Fisher Scientific). Internal standard peak areas were then extracted manually using Quan Browser (Thermo Fisher Xcalibur ver. 2.7), normalized to weight and internal standard peak area, then graphed using GraphPad PRISM (ver 9.0).

**Golgi-Cox staining.** Cryosectioned coronal brain sections (150 μm) were stained using an FD Rapid GolgiStain™ Kit (FD NeuroTechnologies, USA) as instructed by the manufacturer. Staining images were obtained with 10x, 20x, 40x, and 100x lens under bright field with an Olympus IX83 epifluorescent microscope and processed with Imaris (Bitplane, Switzerland) using semi-automated measurements. Z-stacked images taken at 20x magnification were used to include all branches for the creation of 3-D images of dendrites and spines, and measurements of dendritic length, branch point, Sholl analysis, and spine density (from the secondary or tertiary dendrites) was conducted with the FilamentTracer module. At least 2–3 neurons were analyzed from each image; three images were taken per area; four brains per group. Spine morphology were manually measured for head width, neck width, and spine length in a blinded manner. With slight modification[86], spines with a head-to-neck ratio > 1.5, with head width > 0.35 nm were counted as mushroom-shaped; spines with a head-to-neck ratio < 1.5, and a length-to-width ratio < 2.5 were counted as the stubby type; all the rest were counted as thin /filopodial spines[86]. A total of 130~200 spines per group were blindly measured from four brains, 2–3 images per brain area, 10–20 dendritic spines per image. Mushroom, stubby, and thin /filopodial spines were counted and expressed as the percentage of all spine counts.

**Transmission electron microscopy (TEM).** Mice were transcardially perfused with 2% paraformaldehyde and 2.5% glutaraldehyde in PBS described previously[71]. After post-fixed for 24 h, brains were sectioned into 2 mm thick slices and medial corpus callosum was dissected from CL or IL hemispheres, respectively. Tissues were washed three times in PBS before postfixed in 1% Osmium Tetroxide with 1% potassium ferricyanide for 1 h. Following three additional PBS washes, the tissue was dehydrated through a graded series of 30–100% ethanol, 100% propylene oxide and then infiltrated in 1:1 mixture of propylene oxide: Polybed 812 (Luft formulations) epoxy resin for 1 hr. After several changes of 100% resin over 24 h, tissue was embedded in a final change of resin, cured at 37 ℃ overnight, followed by additional hardening at 65 ℃ for two more days. Ultrathin (70 nm) sections cut on a Leica Reichart Ultracut (Leica Microsystems, Buffalo Grove, IL) were collected on 200 mesh copper grids, stained with 2% uranyl acetate in 50% methanol for 10 min, followed by 1% lead citrate for 7 min. Sections were imaged using a JEOL JEM 1400plus transmission electron microscope (Peabody, MA) at 80 kV fitted with a side mount AMT 2k digital camera (Advanced Microscopy Techniques, Danvers, MA). G-ratio calculations were done on 10,000× images from three animals per genotype. G-ratios (inner axon diameter/total diameter) were measured on ~270 myelinated axons per group (three biological independent animals per group, three images per animal, 30 myelinated axons per image) using ImageJ software. Slopes of the linear regressions were compared to identify the differences between groups.

**Immunofluorescent staining.** Mice were transcardially perfused with 0.1 M PBS (pH 7.4), followed by ice-cold 4 % PFA in 0.1 M PBS as described before[18]. Brains were cryoprotected with 30% sucrose after an overnight post-fixation in 4% PFA[18]. Coronal sections (25 μm thickness) were sectioned using a Leica SM2010R microtome (Leica, Germany) for immunofluorescent staining. The sections were incubated with blocking solution (10% normal goat serum and 0.3% Triton X-100 in PBS) and mouse on mouse (M.O.M.) kit (Vector Laboratories, USA) respectively, before incubating with the following antibodies for overnight at 4 ℃: rabbit

polyclonal anti-MBP (1:200, Abcam, USA); mouse monoclonal anti-Olig2 (1:200, Millipore, USA) and rabbit polyclonal anti-NG2 (1:200, Millipore, USA) or rabbit polyclonal anti-Ki67 (1:200, Millipore, USA) or rabbit polyclonal anti-caspase3 (1:200, Cell Signaling Technology, USA) or rabbit polyclonal anti-H3K9me3 (1:200, Abcam, USA); mouse monoclonal anti-PSD95 (1:500, Cell Signaling Technology, USA) and rabbit polyclonal anti-C1q (1:100, Cell Signaling Technology, USA); mouse monoclonal anti-SMI32 (1:200, BioLegend, USA) and rabbit polyclonal anti-APP (1:200, Cell Signaling Technology, USA). After washing in TBS with 0.3% Triton X-100 for 3×10 min, the sections were incubated with goat antimouse Alexa 546-conjugated IgG (1:200, Thermo Fisher Scientific) and goat antirabbit Alexa 488-conjugated IgG (1:200, Thermo Fisher Scientific), or goat antimouse Alexa 488-conjugated IgG (1:200, Invitrogen) and goat antirabbit Alexa 546-conjugated IgG (1:200, Invitrogen) for 1 h. For negative controls, brain sections were stained with the secondary antibodies only. Nuclei were stained with To-pro-3 (1:500, Thermo Fisher Scientific) and sections were mounted with Vectashield mounting medium (Vector Laboratories). Under 40× objective lens using an Olympus IX81 confocal microscope (Olympus, Japan), fluorescent images were captured in peri-lesion areas which were defined as 150-450 μm from the margin of ischemic core showing condensed nuclei or tissue losses. Identical digital imaging acquisition parameters were used and images were obtained and analyzed in a blinded manner throughout the study.

**Statistics and reproducibility**. Unbiased study design with randomized allocation and blinded analyses were implemented in all experiments. Blinding of investigators to experimental groups were maintained until data were fully analyzed whenever possible. Power analysis was performed with 80% power and α (two-sided)=0.05 to detect 20% changes. All measurements were taken from biological replicates unless specified otherwise. Data were expressed as mean ± SEM (GraphPad Prism, USA). Normal distribution were tested and two-tailed Student's $t$-test with 95% confidence was used when comparing two conditions. For more than two conditions, one-way or two-way ANOVA analysis was used, depending on the data. Nonparametric data were analyzed with Mann–Whitney Test. Correlation analysis was performed with Pearson correlation coefficient. $P$ value < 0.05 was considered statistically significant. A total of 212 mice (male and female) were used in the study. All data were included unless appropriate outlier analysis suggested otherwise.

**Reporting summary**. Further information on research design is available in the Nature Research Reporting Summary linked to this article.

## Data availability

All the data associated with this study are present in the paper or the Supplementary Materials. Dataset is provided in Supplementary Data 1. The RNA sequencing data have been deposited to the Gene Expression Omnibus database with experiment series accession number GSE175504.

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

## Acknowledgements

This study was funded by National Institutes of Health R01NS048216 (DS), American Heart Association 17POST32440002 (SS), National Institutes of Health S10OD023402 (SGW), and National Institutes of Health S10OD016236 (DBS). We thank the Molecular Biology Information Service of the Health Sciences Library System at the University of Pittsburgh to provide licenses for Partek Flow and Ingenuity Pathway Analysis software, and the University of Pittsburgh Center for Research Computing (CRC) for the resources in conducting the RNAseq analysis.

## Author contributions

Conceptualization: S.S., D.S. Methodology: S.S., L.Y., M.N.H., S.S.P., S.J.M., M.L.G.S., V.M.F., C.B.Y., D.G.S., S.G.W. Investigation: S.S., L.Y., M.N.H., S.J.M., M.L.G.S., D.G.S., S.G.W., D.S. Visualization: S.S., L.Y., M.N.H., S.J.M., M.L.G.S., D.G.S., S.G.W., D.S. Funding acquisition: D.S., S.S., S.G.W., D.G.S. Project administration: D.S. Supervision: D.S. Writing – original draft: S.S., D.S. Writing – review & editing: S.S., M.N.H., S.J.M., M.L.G.S., D.G.S., S.G.W., D.S.

## Competing interests

The authors declare no competing interests.
