## [Peer Review File · Communications Biology]

Reviewers' comments:

Reviewer #1 (Remarks to the Author):

The manuscript presents further observations obtained using a mouse with conditional, microglia-specific deletion of the NHE1 sodium/proton exchanger, in follow-up to the authors' prior publication in *Glia*. This is a very extensive study, results of which will be of interest to the stroke field. The studies are very nicely performed and presented, showing numerous differences in the microglia derived from wt vs, NHE1 ko mice as well as improved outcomes in the NHE1 ko mice after MCAo. A limitation to the study is that it does not provide further mechanistic insight into how NHE1 deletion leads to the outcomes described, but the study is nevertheless a substantive contribution to the literature. Please see below for some specific comments pertaining to methods and data interpretation.

Specific comments:

- 1) The title, the final two sentences of the abstract, and the conclusions section all attribute causal relationships between the observed changes in metabolism and other outcome measures, whereas only correlations are demonstrated. These sections should all be revised accordingly. Likewise, the emphasis on glucose metabolism in these places seems unfounded, given that large changes in OXPHOS metabolism are demonstrated.
- 2) Results of the histological observations presented in Figs 4 and 6 will be enormously influenced by distance between where the images were taken and the infarct lesion edge. The methods do not address this critical issue, and without that these data are not interpretable.
- 3) Knockdown of NHE1 is obviously not a therapeutic strategy, and so it would be useful to have more insight into the mechanisms by which NHE1 deletion mediates the several effects described. Are they all attributable to changes in intracellular pH? If so, by what mechanism(s)? To be fair, these issues are likely beyond the scope of the present study, but nevertheless deserve some discussion.
- 4) A critical aspect for the interpretation of this study is documentation that NHE 1 expression is down regulated in the microglia and not affected in any other cell type. This was documented in the *Glia* paper, it may be helpful to more clearly reference that work here.
- 5) The methods state that an interval of time is allowed to elapse between tamoxifen administration and MCAo, in order to allow replenishment of normal circulating monocytes. Some aspects of this may benefit from clarification in the text:
 - a) Is this because monocytes also express Cx3cr1? Do other cell types also express Cx3cr1?
 - b) The literature suggests an ongoing trafficking between peripheral monocytes and brain microglia. How does this affect interpretation of the results?
 - c) The literature also suggests a major role for circulating immune cells in both phagocytosis of damaged brain cells and trophic effects after ischemia-reperfusion. This literature should be addressed in the discussion of the present findings.
- 6) As discussed on page 9, the fraction of microglia that take up bioparticles increases from 87% in wt brains to nearly 100% in the cKO brains. The methods section indicates that this analysis was performed using whole ipsilateral hemispheres from ischemic brains, but only a small fraction of the hemisphere was ischemic. This requires some comment.
- 7) The manuscript emphasizes genes that are upregulated in the cKO vs wt microglia, but the data shown in Fig 2 shows a much larger effect on genes that are downregulated. This requires some discussion.
- 8) Fig 5 – why is there no ctrl cKO group in Fig 5f-g?
- 9) Why are no controls shown in Fig S5? Also in Fig S5, why are results here observed in both hemispheres, in contrast to most of the other data shown?
- 10) Fig 1C and Fig 1E show performance of the stroke cKO mice to be better even than the non-stroke wt mice. Presumably these are just statistical outliers, to be expected given the many outcome measures analyzed, but nevertheless deserves some comment in the text.

Reviewer #2 (Remarks to the Author):

Song et al. present a novel characterization of NHE1 functions using mice in which CX3CR1+

microglia are selectively deficient in NHE1, and they investigate outcomes on stroke pathology. Their findings reveal that NHE1 has a direct influence on microglial glucose metabolism, with NHE1 floxed microglia having an increased glucose metabolism. They showed strong brain effects when NHE1 was eliminated from microglia: better post-stroke behavioral outcomes, increased axonal stability, reduced axonal damage, as well as preserved dendritic spines after stroke. They also showed that NHE1 depleted microglia were more phagocytic and presented more contacts with spines. Overall, the study is well designed and conducted, while the findings are very interesting, providing novel data, within this exciting 'immunometabolism' field. I appreciated the depth of characterization Song and colleagues have performed. The model shows striking effects, and the study provides new knowledge regarding microglial functions. Nevertheless, three main limitations were noted, as detailed below, and several suggestions were made to increase further the relevance of the findings to the field.

A first major concern pertains to the use of P2YR12 as a marker to isolate microglia for RNAseq. As mentioned also below, P2RY12 is a marker of homeostatic microglia, which has been reported to be downregulated in aged microglia or in microglia from disease mouse models. My concern is that by targeting P2YR12+ microglia, a subset of microglia found in healthier regions would be selectively isolated, which could be misleading. It would be important to demonstrate that in a stroke context microglial P2YR12 expression is stable. Performing staining against P2YR12 specially in the stroke area would allow to confirm its presence in microglia. Otherwise, the RNA results could be representative of only a fraction of all microglia. A second major concern also explained further below pertains to the interpretation of the results surrounding microglial involvement in the phagocytosis of synapses. The use of C1q marker is not sufficient to conclude that synapses are being phagocytosed by microglia. C1q plays broad roles, complement mediated and non-mediated, notably in contexts of inflammation including but not exclusive to synaptic pruning. Markers of lysosomal or phagolysosomal activity would need to be used. A third point pertains to the comparison with DAMs. While the description of DAMs has been highly valuable to the field, as the microglial discoveries go, comparing everything to DAMs will limit our understanding of the specific response that microglia have to specific disease contexts. I would suggest expanding more on the differences observed between NHE1 cKO contralateral and ipsilateral microglia as well.

Abstract

pHi should be defined for non-specialists

Introduction

Page4, Line 3: this sentence should be revised grammatically

Page 4, Line 3: The microglia field is shifting away from the classically activated pro-inflammatory/ alternatively activated anti-inflammatory phenotypes. These terms originated from peripheral immune studies mainly conducted in vitro and there is a growing body of evidence showing that the on/off states of inflammation/anti-inflammatory phenotypes do not represent microglial responses. Inflammatory processes can be regenerative and anti-inflammatory signals can also slow down repair. I would suggest mentioning them as repairing or damaging states as they can reflect a diversity of microglial responses.

Page4, Lines 4-5: the activated nomenclature should be revised throughout the manuscript considering that microglia are always active, in health and disease

Page 4, Line 5: This works as a continuation to the previous point. This statement might be misleading the reader to think that inflammation is equivalent to dysfunction and anti-inflammatory states a synonym of regulation. In several studies that address disease contexts it is not clear if the inflammatory process is positive or negative. I would rephrase to avoid this confusion.

Page 4, Line 9: This statement is misleading, microglia are constantly surveilling the brain, this sentence makes it seem that surveillance and phagocytosis are events that happen upon activation only. Microglial surveillance and phagocytosis also occur without insult.

Page4, Line 12: similarly, microglia are no more considered to be quiescent or resting in health (vs activated in disease). It would be best to use the surveillant or homeostatic wordings here.

Page4, Lines 19-22: considering that microglial functions are context dependent, I would recommend adding information on the particular context where these findings were obtained

Page 4, Line 22: Microglial clearance defects also impair remyelination (e.g. doi:

10.1084/jem.20141656). This information could be added to differentiate between increased damage and a repairing process.

Page 4, Line 23: a distinction must be made between synaptic pruning via phagocytosis and synaptic stripping, which refers to the physical separation between pre- and post-synaptic elements by intervening microglial processes.

Page 5, Line 5: pHi should be defined for non-specialists

Page 5, Lines 4-7: it would be important to mention whether the findings were obtained in vitro or ex vivo, and to provide more information regarding the samples (cells, specie, brain region, etc.)

Page 5, Line 13: RNAseq should be defined

Page 5, Line 16: LXR and RXR should be defined

Page 5, Lines 17-21: more information should be provided regarding the model used and how the observations were made, to strengthen the conclusion (Lines 21-22)

For increased clarity and translational relevance, throughout the Introduction, it would be important to mention whether the findings were observed in animal models (if so, of which specie) or in human.

Results

Page 6, Line 10: Are these the only sex differences observed in the whole study? If the results on males and females were not significantly different, I would suggest adding a few lines saying that no significant difference were found, and that males and female mice were pooled together.

Page 7, Line 1: Would it be possible to confirm or add to the supplementary the open field test from the WT control mice? WT stroke mice are shown as hyperactive compared to the NHE1 cKO but how does their activity compare to the WT controls?

Page 7, Line 12: the rationale for selecting P2RY12 for microglial isolation should be explained. P2RY12 is a homeostatic marker generally down regulated in contexts of disease including stroke. It is thus expected that microglia nearby the stroke foyer would not be isolated using this marker.

Page 8, Line 19: it would be important to also discuss the MgND associated with contexts of neurodegeneration and other microglial subsets sharing markers with the DAMs

Page 8, Line 23: There is a growing body of evidence showing that developmental microglia and early-stage microglia have a distinct gene expression signature compared to adult microglia. I would recommend keeping in mind these differences to avoid possibly misleading comparisons.

Page 9, Line 1: I would recommend removing weight on the resemblance with DAM cells considering that the investigated models are very different. In the long-term this could limit the impact of the presented work. The current work does a really good job at describing the effect that NHE1 has in a stroke model and in my opinion trying to relate it to the DAM cells does not add more. As a side note the work done on DAM cells has been a valuable addition to the field, the point here is that as microglial discussion goes, comparing everything to DAM cells might limit our understanding of the specific response that microglia have to additional disease contexts. I would also suggest expanding more on the differences observed between NHE1 cKO contralateral and ipsilateral microglia.

Page 9, Line 3: the wording activated microglia-like cells is unclear

Page 9, Line 15: I would discuss the limitations of the in vitro studies as this adds an extra layer of complexity. The results are interesting while acknowledging differences between in vivo and in vitro studies would be important.

Page 9, Line 22: Importantly, C1q plays broad roles, complement mediated and non-mediated, notably in contexts of inflammation including but not exclusive to synaptic pruning. The wording used in the Results should be modified accordingly.

Page 10, Line 4, it is unclear whether microglial expression of C1q is indicative of synaptic pruning via phagocytosis. Co-labeling with a marker of lysosomes (e.g. LAMP1) or phagolysosomal activity (e.g. CD68) would be required.

Page 10, Line 6: synapses should be reworded 'dendritic spines' for accuracy here and elsewhere in the manuscript considering that PSD95 only was used as a synaptic marker.

Page 10, Line 15: 'reduction in spine density was also observed on dendrites' should read

Page 11, Lines 20-21: the CC and EC could be linked to the behavioral deficits observed post-stroke for additional relevance

Page 12, Line 6: I would recommend providing more insight into what the difference between large and medium axon size could mean

The titles of the results sections could be rephrased to better represent the coming findings

Discussion

Page 14, Line 3: I would also include the actual digestion of the elements being phagocytosed

Page 14, Line 19: the term 'identical' is too strong here, as mentioned previously, comparing the microglial states observed in this study to DAMs is interesting but could be an important limitation. Microglial states are context-dependent and as much, most likely not identical across different disease conditions. These states tightly depend on the type of injury, timing, brain region, sex, age, etc.

Page 15, Line 9: the alternatively activated phenotype wording should be modified here, considering that the M1 and M2 polarization has been rejected by the field

Page 15, Lines 18-21: the sentence is unclear, as written with the subject microglial phagocytic function, especially how this function is linked to BDNF promotion of dendritic spine formation. The discussion about phagocytosis of damaged synapses or spines should be revised, unless a marker of lysosomes or phagolysosomal activity is used, as mentioned above

Page 16, Line 1: the apoptotic-like synapses should be explained further, and references provided

Page 16, Line 12: The way this is written makes me think that the hyperactivity observed in the WT post-stroke is a result of memory differences, to revise

Conclusion

The first sentence of the conclusion appears overstated and should be revised

Figure 4

The wording related to the phagocytosis of synapses should be revised as mentioned previously

Reviewer #3 (Remarks to the Author):

This paper revealed that genetic blockade of microglial NHE1 stimulated glucose immunometabolism and phagocytic function for tissue remodeling and post-stroke cognitive function recovery.

The manuscript was well written and the experiments generally supported the author's hypothesis. There are a few minor comments for the authors.

Minor comments

1. Cognitive abilities after stroke in WT measuring by Y maze and NOR were not different from those in non-injured mice (Fig 1C and E). According to the authors' previous report published in *Glia*, neurological scores in WT did not recover well until 14 days after MCAO. Can cognitive function be fully restored 28 days after stroke?
2. The number of genes upregulated in cKO is 3349, which is more than 20 times that of WT (Fig2 E). Is there any specific reason? Need an explanation.
3. Authors have used anti-P2RY12 to isolate microglia. P2RY12 is one of the homeostatic markers. There are some papers showing P2RY12 mRNA are reduced by microglia activation (such as LPS stimulation). Ref : P2RY12 mRNA is reduced by IFN γ +LPS in microglia from WM and GM (<https://actaneurocomms.biomedcentral.com/articles/10.1186/s40478-019-0850-z>). It would be good to have discussion on it.
4. The balance between Glycolysis and OXPHOS is important for microglial function as authors mentioned in introduction. It is not clear what the status of cKO microglia is because overall glucose metabolism is upregulated. Can upregulated glucose metabolism be beneficial? Are there any references indicating a condition similar to cKO microglia?
5. In Fig.4H, branch points and dendrite length are significantly higher in cKO brain compared to WT even in non-injured brain.
6. Authors showed significantly enhanced myelination of CC in the cKO brains (midline thickness at the same bregma level, Supplementary Figure 4C): How about CC thickness in non-injured brain?
7. Fig4 D. colocalization images are from IL tissues or isolated microglia. Please make a clear in fig legend
8. Fig4H, Add information on the replicates of neurons as well as brains or experiments.
9. In Fig. 4B and 4E showing quantification of microglial phagocytic activity (N=4). What N means? Animals, cells or experiments? Need information on how many cells and animals were used for the graph (for all figure legends).

We would like to thank our reviewers for their critical review and many helpful suggestions. In addressing the reviewers' concerns, we conducted new experiments and included new data in **Figures 4-5**, and **Supplementary Figures 1-7 and 9**, which further strengthened our conclusion about coordinated regulation of microglial energy metabolism in support of phagocytic functions. We have thoroughly revised the manuscript according to the reviewers' suggestions, and the manuscript has been significantly improved. The followings are our point-by-point responses.

Reviewer #1

1. The title, the final two sentences of the abstract, and the conclusions section all attribute causal relationships between the observed changes in metabolism and other outcome measures, whereas only correlations are demonstrated. These sections should all be revised accordingly. Likewise, the emphasis on glucose metabolism in these places seems unfounded, given that large changes in OXPHOS metabolism are demonstrated.

Response: As suggested by the reviewer, we revised the title as: "Stimulating microglial oxidative phosphorylation and phagocytosis for post-stroke remyelination and cognitive function recovery". To address Reviewer's concerns on significance of microglial glycolysis and OXPHOS, we performed additional Seahorse assays. Our new data showed that the cKO microglia exhibited reduced glycolysis but stimulated OXPHOS capacity after stroke (please see **Fig. 5H-J**). With these findings, we revised our abstract in **page 3, line 10-15** as: "The cKO microglia exhibited increased OXPHOS capacity, and concurrently higher phagocytic activity, which likely played a role in enhanced synaptic stripping and remodeling, oligodendrogenesis, and remyelination. This study reveals that genetic blockade of microglial NHE1 stimulated OXPHOS immunometabolism, and boosted phagocytosis function for tissue remodeling and post-stroke cognitive function recovery".

The conclusion section in **page 23, line 14-18** has been revised as: "In this study, we discovered that transgenic deletion of the major microglial pH-regulating protein NHE1 switched microglial metabolism from glycolysis to OXPHOS to generate more ATP. These microglia concurrently showed elevated phagocytosis activity, and enhanced synaptic remodeling and myelin repair. Improved post-stroke cognitive function recovery was observed following these metabolic alterations (**Figure 8**)."

2. Results of the histological observations presented in Figs 4 and 6 will be enormously influenced by distance between where the images were taken and the infarct lesion edge. The methods do not address this critical issue, and without that these data are not interpretable.

Response: As suggested by the reviewer, we included illustrations of the areas where histological images were taken in both **Figs. 4 and 6**, and revised the corresponding figure legends. In page 39, line 8-9 “Representative staining images of C1q, PSD95, and To-pro-3 from the IL peri-lesion area at 3 d post-stroke”; in page 40, line 2 “Representative 40x Golgi-Cox staining images of layer V neurons in the IL peri-lesion cortex of WT and cKO brains at 28 d post-stroke”; in page 43, line 4-5 “Representative images and g-ratio from axons in the medial corpus callosum (CC) by transmission electron microscopy (TEM)”; in page 43, line 7-9 “Representative images and quantitative analysis of SMI32 and APP intensity in CL and IL peri-lesion areas of CC from WT or *Nhe1* cKO mice at 3 d post-stroke”.

3. Knockdown of NHE1 is obviously not a therapeutic strategy, and so it would be useful to have more insight into the mechanisms by which NHE1 deletion mediates the several effects described. Are they all attributable to changes in intracellular pH? If so, by what mechanism(s)? To be fair, these issues are likely beyond the scope of the present study, but nevertheless deserve some discussion.

Response: Regarding the possible mechanisms underlying NHE1 protein involvement in microglial transcriptome and energy metabolism, we speculate that NHE1 plays the modulatory role via regulating microglial pH_i homeostasis. Specifically, transgenic *Nhe1* deletion resulted in inhibition of the NHE1-mediated H⁺ extrusion, thus, preventing microglial intracellular alkalization and Warburg effect on stimulating glycolysis, the latter mechanism has been well established ¹. This view is further supported by our new experiments with biogenetic analysis of oxygen consumption rate (OCR) and extracellular acidification rate (ECAR) with the Seahorse Extracellular Flux Analyzer. We have included a new result section on page 13, line 17-page 14, line 9: “Moreover, **Fig. 5H-J** showed that microglial cells from the non-lesion CL hemispheres displayed low basal glycolysis-mediated extracellular acidification, an indirect analysis of the glycolytic rate of cells². Stroke triggered significant stimulation of glycolysis in WT microglia, reflected by the increased basal glycolysis as well as glycolytic capacity (**Fig. 5H-I**). In contrast, cKO microglial cells exhibited significantly slower ECAR, indicating less H⁺ extrusion rate and reduced glycolysis (**Fig. 5H-I**). Overall, the bioenergetics profile of WT and cKO microglia in **Fig. 5J** clearly demonstrated that microglia from the non-lesion CL brain tissues are less energetic with low respiration and glycolysis. WT microglia of stroke brains significantly ramped up their glycolytic activity than mitochondrial OXPHOS and thus showed more glycolytic phenotype. In contrast, the cKO microglia of stroke brains relied more on the OXPHOS for their energy demand (**Fig. 5J**). These data suggest that reduced glycolysis in the cKO microglia after stroke likely

resulted in less NHE1-mediated H⁺ extrusion and more acidic pH_i, preventing Warburg effect on stimulating glycolysis¹. These data strongly suggest that loss of microglial NHE1 protein plays an important role in boosting their OXPHOS metabolism and ATP production to provide fuels in supporting microglial phagocytic functions in the *Nhe1* cKO brains.”

In addition, the following discussion has been included in page 22, line 22-page 23, line 5: “Of note, pH_i homeostasis is involved in regulating energy metabolism as well as transcriptome changes³. More acidic pH_i has been linked to stimulating the master transcription regulator *Hif1a*³. Whether these biochemical mechanisms play a role in the transcriptome changes in cKO microglia is warranted for further study. In addition, pharmacological approaches using potent NHE1 inhibitors such as Cariporide or Rimeporide⁴⁻⁶ should be employed to assess their efficacy in stimulating microglial energy metabolism and phagocytosis in post-stroke brain repair.”

4. A critical aspect for the interpretation of this study is documentation that NHE1 expression is down regulated in the microglia and not affected in any other cell type. This was documented in the *Glia* paper, it may be helpful to more clearly reference that work here.

Response: As suggested by the reviewer, we described our previous work in page 6, line 6-7: “We previously reported that *Cx3cr1-Cre*^{ER+/-};*Nhe1*^{ff} mouse line successfully deleted NHE1 protein expression only in IBA1⁺ microglia, but remained intact in other cell types⁷.”

5. The methods state that an interval of time is allowed to elapse between tamoxifen administration and MCAo, in order to allow replenishment of normal circulating monocytes. Some aspects of this may benefit from clarification in the text:

a) Is this because monocytes also express *Cx3cr1*? Do other cell types also express *Cx3cr1*?

Response: As suggested by the reviewer, we included additional clarification in **Supplementary Materials and Methods**, page 1, line 13-19: “As *Cx3cr1* is expressed by both brain resident microglia and peripheral infiltrating bone marrow-derived myeloid cells (BMDM), a 30-day post-injection waiting period was given for clearance of Tam⁸⁻¹⁰ and for replenishing of *Cx3cr1*⁺ monocytes¹¹ prior to induction of ischemic stroke (**Fig. 1A**), considering the *Cx3cr1*⁺ BMDM, but not the *Cx3cr1*⁺ microglia, have a 21-day self-renewing cycle¹². This method has been proven to achieve a complete replenishment of *Cx3cr1*⁺ BMDM¹¹ and is effective in our study^{7,13} as well as in others¹⁴⁻¹⁶.”

b) The literature suggests an ongoing trafficking between peripheral monocytes and brain microglia. How does this affect interpretation of the results?

Response: To address the reviewer's concerns, we performed new flow cytometry analysis to determine changes of brain infiltrated immune cells and splenic immune cells (the main peripheral source of immune cells for infiltration) in WT and cKO mice. The following paragraph has been included in **page 12, line 11-page 13, line 3**: "Considering the communications between microglia and infiltrated peripheral monocytes¹⁷, we further assessed changes of macrophages, neutrophil, and T cells in the spleen and their brain infiltration in WT and cKO mice with flow cytometry (**Supplementary Fig. 5A**). **Supplementary Fig. 5B** showed that similar cell counts of spleen CD11b⁺CD45^{hi} macrophages, CD11b⁺CD45^{hi}Ly6G⁺ neutrophils, and CD3⁺T cells were detected in WT and cKO sham mice. At 3 d post-stroke, no significant changes of spleen immune cells were detected in WT or cKO mice. These findings are consistent with reports that stroke did not significantly change the number of macrophages and neutrophils in the spleen¹⁸. In contrast, **Supplementary Fig. 5C** showed that stroke led to an increased brain infiltration of macrophages, neutrophils, and T cells in WT as well as cKO mice, similar to other reports of stroke brains^{18,19}. Taken together, our data imply that selective deletion of *Nhe1* in microglial cells did not significantly change either immune cell infiltration in stroke brains or immune responses in the spleen."

c) The literature also suggests a major role for circulating immune cells in both phagocytosis of damaged brain cells and trophic effects after ischemia-reperfusion. This literature should be addressed in the discussion of the present findings.

Response: As suggested by the reviewer, we added discussion in **page 17, line 6-9**: "It is worth mentioning that circulating monocytes can also become phagocytic and beneficial for debris clearance upon ischemic injury¹⁷, despite that we detected no differences in CD11b⁺CD45^{high} macrophage cell counts between the WT and cKO brains through 3-14 d post-stroke⁷." Please also see Question 5b for our new characterizations of brain infiltrated and splenic immune cells.

6. As discussed on page 9, the fraction of microglia that take up bioparticles increases from 87% in wt brains to nearly 100% in the cKO brains. The methods section indicates that this analysis was performed using whole ipsilateral hemispheres from ischemic brains, but only a small fraction of the hemisphere was ischemic. This requires some comment.

Response: As suggested by the reviewer, we added sentences in **page 10, line 10-13**: "We previously reported that the infarct volume was similar between the WT and cKO post-stroke mice⁷, therefore we assessed whether the microglial phagocytic activity in hemispheric brain tissues differs in WT and cKO brains."

In addition, we included the microglial phagocytosis in the CL hemispheres in the revised **Fig. 4B**, We described the results in **page 10, line 16-18**: “In addition, microglial phagocytic activity in the non-stroke CL hemisphere from cKO brains was also significantly increased compared to the WT brains (**Figure 4B**).” These data suggest that deletion of *Nhe1* in microglia increased microglia phagocytosis in both lesioned and non-lesion hemispheres, which may lead to the enhanced synaptic remodeling in both hemispheres at a later resolving phase as shown in **Fig. 4F-M**.

7. The manuscript emphasizes genes that are upregulated in the cKO vs wt microglia, but the data shown in Fig 2 shows a much larger effect on genes that are downregulated. This requires some discussion.

Response: As suggested by the reviewer, we added discussion in **page 18, line 9-14**: “Moreover, the post-stroke cKO microglia displayed a large number of downregulated genes compared to its non-lesion hemisphere, or to the post-stroke WT microglia (**Figure 2D-E**). IPA pathway analysis of the DEGs revealed the downregulated pathways were mostly involved in inflammatory responses, such as the OX40 signaling pathway (**Figure 3A**), which is involved in NF-kB activation²⁰, consistent with previous report that NHE1 blockade inhibited NF-kB activation²¹”.

8. Fig 5 – why is there no ctrl cKO group in Fig 5f-g?

Response: The metabolism assay using Seahorse Extracellular Flux Analyzer requires a large number of microglial cells. We normally need to pool microglial cells from two naïve mice to conduct an experiment as shown in **Supplementary Fig. 6** (for WT and cKO mice). The non-stroke CL hemispheres possessed similarly low microglial cell counts. We did not detect differences of microglial glycolytic metabolism between naïve WT and cKO mice (**Supplementary Fig. 6**). Therefore, we pooled the P2RY12+ cells from the CL hemispheres of WT and cKO stroke mice as the control group in **Fig.5**. We included these results in **page 13, line 8-11**, and clarified in the figure legend in **page 43, line 3-5** as: “As we observed similar results in non-lesion CL hemispheres from WT and cKO brains, these samples were pooled to represent as the non-lesion control (Ctrl)”.

9. Why are no controls shown in Fig S5? Also in Fig S5, why are results here observed in both hemispheres, in contrast to most of the other data shown?

Response: We have shown both CL and IL data in our histological image analysis, including **Fig. 4H-M, Fig. 6C, and Fig. 7A-D**. For **Fig. S8** (previously **Fig. S5**), we used the non-stroke CL hemisphere as the internal control, which controls for individual and environmental variances.

10. Fig 1C and Fig 1E show performance of the stroke cKO mice to be better even than the non-stroke wt mice. Presumably these are just statistical outliers, to be expected given the many outcome measures analyzed, but nevertheless deserves some comment in the text.

Response: Outliers were identified if the data points were below or above 1.5 times the interquartile range (IQR) below the first quartile or above the third quartile. By this definition, the two data points in post-stroke cKO group were not outliers [25 percentile = 63.69; 75 percentile = 83.89; IQR = 20.2; 1.5*IQR = 30.3. Therefore, only data below 33.39 (63.69-30.3) or above 114.19 (83.89+30.3) were outliers].

We also added the following discussion in page 20, line 6-15: “Regarding the better performance of cKO mice post-stroke than post-sham in the y-maze test, the possible causes are not clear. Previous report has shown that stroke in the prefrontal cortex could facilitate an improvement in learning ability in aged mice, which outperformed their aged sham controls²². This improvement in cognitive flexibility may lie in the neural circuitry enhancement and was proven to be dependent on an elevation in BDNF levels in these post-stroke animals²². Our data demonstrated that the cKO stroke brains indeed have improved synaptic remodeling for network strengthening, which could contribute to their better performance. Future studies are warranted to examine whether changes of the BDNF levels in these cKO microglia contribute to an enhancement of the neural circuit in cKO mice.”

Reviewer #2

1. A first major concern pertains to the use of P2YR12 as a marker to isolate microglia for RNAseq. As mentioned also below, P2RY12 is a marker of homeostatic microglia, which has been reported to be downregulated in aged microglia or in microglia from disease mouse models. My concern is that by targeting P2YR12+ microglia, a subset of microglia found in healthier regions would be selectively isolated, which could be misleading. It would be important to demonstrate that in a stroke context microglial P2YR12 expression is stable. Performing staining against P2YR12 specially in the stroke area would allow to confirm its presence in microglia. Otherwise, the RNA results could be representative of only a fraction of all microglia.

Response: **Supplementary Figure 2D** showed that our bulk RNAseq detected similar level of *P2ry12* gene expression in WT or cKO microglia isolated from stroke brains (**Supplementary Figure 2D**). Moreover, flow cytometry data showed that the CD11b⁺CD45⁺P2RY12⁺ microglial population remained comparable between the WT and cKO brains post-stroke (CL and IL hemispheres, **Supplementary Figure 2D**). The CD11b⁺CD45⁺P2RY12⁺ microglial population

displayed significantly higher phagocytosis activity (**Figure 4B**). These findings collectively suggest that the isolated P2RY12⁺ microglial population from stroke brains represent homeostatic as well as polarized microglia and that it is unlikely that we selectively only assessed the homeostatic microglial population. This view is supported by a recent report²³ that P2RY12 protein remained abundant and detectable in proinflammatory, phagocytosing microglia in AD brains, where P2RY12 microglial cells were positive for CD68, progranulin and HLA-DR²³. Since this is an important point for readers, the above discussion has been included in page 7, line 22- page 8, line 8 of the revised manuscript. We attempted to perform immunostaining study for P2RY12 protein, unfortunately, our P2RY12 antibody did not generate specific binding signals, and is not reliable for addressing this issue.

2. A second major concern also explained further below pertains to the interpretation of the results surrounding microglial involvement in the phagocytosis of synapses. The use of C1q marker is not sufficient to conclude that synapses are being phagocytosed by microglia. C1q plays broad roles, complement mediated and non-mediated, notably in contexts of inflammation including but not exclusive to synaptic pruning. Markers of lysosomal or phagolysosomal activity would need to be used.

Response: As suggested by the reviewer, we have performed immunostaining using a lysosomal marker LAMP1, and PSD95 to assess whether the targeted PSD95⁺ synapses has been engulfed/digested in the cellular lysosomes. As shown in **Supplementary Figure 4**, the inclusion of PSD95⁺ synapses within the LAMP1⁺ cells were significantly elevated in cKO stroke brains, compared to the WT brains. This is consistent with the increased microglial phagocytic activity as well as the enhanced synaptic remodeling in the cKO brains detected with the increased C1q/PSD95 labeling, as well as flow cytometry and Golg-Cox staining data. This new information has been included in page 11, line 11-16.

3. A third point pertains to the comparison with DAMs. While the description of DAMs has been highly valuable to the field, as the microglial discoveries go, comparing everything to DAMs will limit our understanding of the specific response that microglia have to specific disease contexts. I would suggest expanding more on the differences observed between NHE1 cKO contralateral and ipsilateral microglia as well.

Response: We thank the reviewer to point out this important aspect. We have included the microglial transcriptome data from the CL hemispheres in **Supplementary Figure 3**, which showed similar results as the IL hemispheres. Therefore, we revised the paragraph in page 9,

line 6-18 as: “Moreover, genes involved in phagocyte recruitment, recognition, and engulfment (**Figure 3D**), as well as in phagolysosomal function and lipid metabolism (related to the digestion process of phagocytosis) were also significantly enhanced in the cKO microglia from both CL (**Supplementary Figure 3**) and IL hemispheres ($p < 0.0001$, **Figure 3E-F**). A recent report identified a new population of microglia in the context of Alzheimer’s disease (AD), termed disease-associated microglia (DAM), with distinct features of elevated phagocytosis, phagolysosomal function, and lipid/cholesterol metabolism²⁴, which shared many molecular signature markers with microglia in neurodegenerative diseases (MGnD)²⁵. However, in our MCAO-induced stroke model, these gene signatures were elevated in both the lesioned IL hemisphere, and the non-lesion CL hemisphere. While microglial responses are largely dependent on specific disease contexts, virtually all of the DAM/MGnD signature genes reported^{24,25} were significantly elevated in the cKO stroke microglia ($p = 0.0001$, **Figure 3G**).”

4. Abstract: pH_i should be defined for non-specialists

Response: As suggested by the reviewer, we defined pH_i in the Abstract: page 3, line 4 “Intracellular pH (pH_i) is important for regulating aerobic glycolysis in microglia...” .

5. Introduction: Page4, Line 3: this sentence should be revised grammatically

Response: As suggested by the reviewer, we revised page 4, line 3 as: “The balance between glycolysis and oxidative phosphorylation (OXPHOS) is important for...” .

6. Page 4, Line 3: The microglia field is shifting away from the classically activated pro-inflammatory/ alternatively activated anti-inflammatory phenotypes. These terms originated from peripheral immune studies mainly conducted in vitro and there is a growing body of evidence showing that the on/off states of inflammation/anti-inflammatory phenotypes do not represent microglial response. Inflammatory processes can be regenerative and anti-inflammatory signals can also slow down repair. I would suggest mentioning them as repairing or damaging states as they can reflect a diversity of microglial response.

Response: As suggested by the reviewer, we revised wording describing microglial responses throughout the manuscript as “damaging” or “restorative”.

7. Page4, Lines 4-5: the activated nomenclature should be revised throughout the manuscript considering that microglia are always active, in health and disease

Response: Please see Question 6 of Reviewer #2. We also avoided using the nomenclature “activated microglia” throughout the text.

8. Page 4, Line 5: This works as a continuation to the previous point. This statement might be misleading the reader to think that inflammation is equivalent to dysfunction and anti-inflammatory states a synonym of regulation. In several studies that address disease contexts it is not clear if the inflammatory process is positive or negative. I would rephrase to avoid this confusion.

Response: Please see responses to Reviewer #2, Question 6.

9. Page 4, Line 9: This statement is misleading, microglia are constantly surveilling the brain, this sentence makes it seem that surveillance and phagocytosis are events that happen upon activation only. Microglial surveillance and phagocytosis also occur without insult.

Response: We revised the sentence in page 4, line 8-11 as: “Microglial cells, as the resident macrophages in the central nervous system (CNS), require a high energy expenditure to support their core functions such as surveillance and phagocytosis, where ATP is in high demand through various glucose metabolism processes²⁶.”

10. Page4, Line 12: similarly, microglia are no more considered to be quiescent or resting in health (vs activated in disease). It would be best to use the surveillant or homeostatic wordings here.

Response: As suggested by the reviewer, we revised page 4, line 11 as: “It is generally believed that homeostatic microglia primarily rely upon OXPHOS for ATP production...” .

11. Page4, Lines 19-22: considering that microglial functions are context dependent, I would recommend adding information on the particular context where these findings were obtained

Response: As suggested by the reviewer, we revised page 4, line 16-20 as: “Microglial phagocytosis is directly involved in efficient clearance of myelin debris in support of white matter remyelination after demyelination injury^{27,28}, while dysfunction of microglial phagocytosis or inadequate elimination of the debris prolonged demyelination and/or impairs remyelination in either stroke or multiple sclerosis mouse models²⁹⁻³¹.”

12. Page 4, Line 22: Microglial clearance defects also impair remyelination (e.g. doi: 10.1084/jem.20141656). This information could be added to differentiate between increased damage and a repairing process.

Response: As suggested by the reviewer, we included the reference paper in page 4, line 20. Bibliography was updated accordingly.

13. Page4, Line 23: a distinction must be made between synaptic pruning via phagocytosis and synaptic stripping, which refers to the physical separation between pre- and post-synaptic elements by intervening microglial processes.

Response: As suggested by the reviewer, we revised the sentence in page 4, line 20-23 as: “In addition, microglial phagocytosis impairment also limited sculpting of neural synapses/networks in synaptic pruning during early development or normal adolescent³², or stripping the weak/injured synapses after ischemic stroke, contributing to cognitive decline in mouse models^{33,34}.”

14. Page 5, Line 5: pHi should be defined for non-specialists

Response: We revised the sentence in page 5, line 4 as: “...is essential in regulating microglial homeostatic intracellular pH (pHi)”.

15. Page 5, Lines 4-7: it would be important to mention whether the findings were obtained in vitro or ex vivo, and to provide more information regarding the samples (cells, specie, brain region, etc.)

Response: As suggested by the reviewer, we revised the paragraph in page 5, line 3-8 as: “We previously reported that Na⁺/H⁺ exchanger isoform-1 (NHE1), which mediates H⁺ efflux in exchange of Na⁺ influx, is essential in regulating microglial homeostatic intracellular pH (pHi) in primary microglial culture³⁵. NHE1-mediated H⁺ extrusion activity alkalinizes microglial pHi (to 7.29 ± 0.02, p < 0.05) and promotes NADPH oxidase (NOX) function upon lipopolysaccharides (LPS) stimulation³⁵. The similar functional link between NHE1 and NOX2 activation has been detected in mouse brains upon NMDA injection or after ischemic stroke³⁶.”

16. Page 5, Line 13: RNAseq should be defined

Response: We revised the sentence in page 5, line 13 as: “...we conducted transcriptomic analysis of post-stroke wild-type (WT) and *Nhe1* cKO microglia by bulk RNA sequencing (RNAseq)”.

17. Page 5, Line 16: LXR and RXR should be defined

Response: We revised the sentence in page 5, line 16-18 as: “Moreover, these cKO microglia showed a panel of upregulated genes for phagocytosis and liver X receptor-retinoid X receptor (LXR/RXR) pathway activation...” .

18. Page 5, Lines 17-21: more information should be provided regarding the model used and how the observations were made, to strengthen the conclusion (Lines 21-22)

Response: As suggested by the reviewer, we revised the paragraph in page 5, line 16-22 as: “Moreover, these cKO microglia showed a panel of upregulated genes for phagocytosis and liver X receptor-retinoid X receptor (LXR/RXR) pathway activation in the Ingenuity Pathways Analysis (IPA), which is also involved in microglial phagocytosis responses³⁷. These changes were corroborated with detection of increased OXPHOS capacity, elevated microglial phagocytic activity of bioparticles, enhanced synaptic remodeling in Golgi-Cox staining, and improved white matter myelination with transmission electron microscopy (TEM) and immunostaining.”

19. For increased clarity and translational relevance, throughout the Introduction, it would be important to mention whether the findings were observed in animal models (if so, of which specie) or in human.

Response: As suggested by the reviewer, we revised the Introduction by including additional information of species and models throughout the paragraphs.

20. Results: Page 6, Line 10: Are these the only sex differences observed in the whole study? If the results on males and females were not significantly different, I would suggest adding a few lines saying that no significant difference were found, and that males and female mice were pooled together.

Response: As suggested by the reviewer, we added a sentence in page 6, line 15-16: “Both sexes showed similar results otherwise, and thus were pooled for analysis.”

21. Page 7, Line 1: Would it be possible to confirm or add to the supplementary the open field test from the WT control mice? WT stroke mice are shown as hyperactive compared to the NHE1 cKO but how does their activity compare to the WT controls?

Response: As suggested by the reviewer, we included representative open field tracing path of WT and cKO sham mice in **Supplementary Figure 1E**. We also revised the sentence in page 7, line 7-10 as: “Moreover, the WT and cKO mice exhibited similar locomotor activity after sham procedures (**Supplementary Figure 1E, Figure 1F-G**), while the post-stroke WT mice displayed

hyperactivity reflected by the increased travel distance compared to the post-stroke cKO mice in the OF test (**Figure 1F-G**).”

22. Page 7, Line 12: the rationale for selecting P2RY12 for microglial isolation should be explained. P2RY12 is a homeostatic marker generally down regulated in contexts of disease including stroke. It is thus expected that microglia nearby the stroke foyer would not be isolated using this marker.

Response: Please see responses to Question 1.

23. Page 8, Line 19: it would be important to also discuss the MgND associated with contexts of neurodegeneration and other microglial subsets sharing markers with the DAMs

Response: Please see responses to Question 3.

24. Page 8, Line 23: There is a growing body of evidence showing that developmental microglia and early-stage microglia have a distinct gene expression signature compared to adult microglia. I would recommend keeping in mind these differences to avoid possibly misleading comparisons.

Response: As suggested by the reviewer, we revised the sentence in page 9, line 18-23 as: “Normal adult microglia altered gene expression signatures during maturation³⁸, however, they can re-express many of the early developmental microglial markers upon focal demyelination injury, which are enriched for pathways associated with cell metabolism, growth, motility, and proliferation³⁸. Interestingly, our cKO microglia also expressed gene markers that were highly resembling the early development microglia (**Supplementary Figure 2E**) with genetic patterns indicative of elevated phagocytosis function³⁸.”

25. Page 9, Line 1: I would recommend removing weight on the resemblance with DAM cells considering that the investigated models are very different. In the long-term this could limit the impact of the presented work. The current work does a really good job at describing the effect that NHE1 has in a stroke model and in my opinion trying to relate it to the DAM cells does not add more. As a side note the work done on DAM cells has been a valuable addition to the field, the point here is that as microglial discussion goes, comparing everything to DAM cells might limit our understanding of the specific response that microglia have to additional disease contexts. I would also suggest expanding more on the differences observed between NHE1 cKO contralateral and ipsilateral microglia.

Response: We have revised this discussion as suggested by the Reviewer. Please see our responses to Question 3.

26. Page 9, Line 3: the wording activated microglia-like cells is unclear

Response: We revised the sentence in page 11, line 9-10 as: "...along with phagocytosing microglial morphology showing increased C1q expression (arrowhead, **Figure 4D**)".

27. Page 9, Line 15: I would discuss the limitations of the in vitro studies as this adds an extra layer of complexity. The results are interesting while acknowledging differences between in vivo and in vitro studies would be important.

Response: As suggested by the reviewer, we revised the paragraph in page 10, line 18-21 as: "These *ex vivo* data suggest that selective deletion of microglial *Nhe1* enhances their phagocytic activity in the non-lesion as well as in the acute stroke tissues, which is in line with our RNAseq transcriptomic findings. However, additional *in vivo* assessment of microglial phagocytic activity is warranted in future study."

28. Page 9, Line 22: Importantly, C1q plays broad roles, complement mediated and non-mediated, notably in contexts of inflammation including but not exclusive to synaptic pruning. The wording used in the Results should be modified accordingly.

Response: As suggested by the reviewer, we revised the sentences in page 11, line 2-6 as: "C1q is the initiating protein of the classical complement cascade, where its cleaved products attract phagocytes and tags target cells for elimination by phagocytosis, thus has been used as a marker for targets of microglial phagocytosis³⁹. Of note, C1q is increased and associated with synapses in the context of Alzheimer's disease⁴⁰".

29. Page 10, Line 4, it is unclear whether microglial expression of C1q is indicative of synaptic pruning via phagocytosis. Co-labeling with a marker of lysosomes (e.g. LAMP1) or phagolysosomal activity (e.g. CD68) would be required.

Response: We conducted new experiments. Please see responses to Question 2.

30. Page 10, Line 6: synapses should be reworded 'dendritic spines' for accuracy here and elsewhere in the manuscript considering that PSD95 only was used as a synaptic marker.

Response: As suggested by the reviewer, we revised the wording in page 11, line 15 and throughout the manuscript.

31. Page 10, Line 15: 'reduction in spine density was also observed on dendrites' should read

Response: We revised the sentence in page 12, line 2 as: “a dramatic reduction in spine density was observed in the WT mice ...” .

32. Page 11, Lines 20-21: the CC and EC could be linked to the behavioral deficits observed post-stroke for additional relevance

Response: As suggested by the reviewer, we revised the sentence in page 14, line 12-14 as: “We previously detected high number of mature oligodendrocytes (APC⁺) in the corpus callosum (CC) and external capsule (EC) of cKO brains at 14 d post-stroke, which correlated with post-stroke functional improvement⁷.”

33. Page 12, Line 6: I would recommend providing more insight into what the difference between large and medium axon size could mean

Response: As suggested by the reviewer, we added a paragraph in page 15, line 1-7: “As reported, in CC of normal adult mice, approximately 60–70% of axons are unmyelinated and these axons do not exceed 0.6 μm in diameter⁴¹. Thus, the greater myelination of small to medium diameter axons could result from myelination of these axons that were not myelinated in normal circumstances⁴¹, which was also correlated with behavioral improvement⁴¹. These data indicate that the cKO mice increased myelination potentially in the previously unmyelinated smaller axons, and provide additional support for the improved neurological functions observed in these cKO mice.”

34. The titles of the results sections could be rephrased to better represent the coming findings

Response: Following Reviewer’s suggestion, we have revised section titles accordingly.

35. Discussion: Page 14, Line 3: I would also include the actual digestion of the elements being phagocytosed

Response: As suggested by the reviewer, we revised the sentence in page 17, line 1-2 as: “We observed that the post-stroke cKO microglia exhibited increased transcriptome profiles for all processes in phagocytosis (recruitment, recognition, engulfment, and digestion)...”

36. Page 14, Line 19: the term ‘identical’ is too strong here, as mentioned previously, comparing the microglial states observed in this study to DAMs is interesting but could be an important limitation. Microglial states are context-dependent and as much, most likely not identical across

different disease conditions. These states tightly depend on the type of injury, timing, brain region, sex, age, etc.

Response: As suggested by the reviewer, we revised the sentence in page 17, line 19 as: “Interestingly, the *Nhe1* cKO microglia showed similar upregulation of these genes after ischemic stroke.”

37. Page 15, Line 9: the alternatively activated phenotype wording should be modified here, considering that the M1 and M2 polarization has been rejected by the field

Response: We have revised the wording throughout the text. Please see responses to Questions 6-8.

38. Page 15, Lines 18-21: the sentence is unclear, as written with the subject microglial phagocytic function, especially how this function is linked to BDNF promotion of dendritic spine formation

Response: We revised the sentences in page 19, line 11-14 as: “Healthy microglial phagocytosis function is required for selectively engulfing weak or injured pre- and post-synaptic elements during development^{42,43}. Microglial secreted trophic factors (such as TGF-beta, BDNF) are also proven important for mediating pruning and formation of dendritic spines and modulating synaptic functions in adulthood^{11,34}.”

39. The discussion about phagocytosis of damaged synapses or spines should be revised, unless a marker of lysosomes or phagolysosomal activity is used, as mentioned above

Response: We have included discussion about LAMP1 staining data. Please see responses to Question 2.

40. Page 16, Line 1: the apoptotic-like synapses should be explained further, and references provided

Response: As suggested by the reviewer, we revised the paragraph in page 19, line 16-23 as: “We detected increased C1q/PSD95 co-localization in the *Nhe1* cKO brains at 3 d post-stroke, where C1q, the initiating protein in the classical complement cascade, can selectively target the apoptotic-like synapses that need to be eliminated by microglia^{39,44,45}. A proteomic study revealed that the C1q-tagged synaptosomes expressed a great number of proteins involved in apoptotic processes, suggesting that synaptic C1q tagging plays a role in the recognition of synapses in which local apoptotic-like processes have been initiated⁴⁴, and thus C1q has emerged as a critical mediator for synaptic refinement and plasticity^{39,44,45}.”

41. Page 16, Line 12: The way this is written makes me think that the hyperactivity observed in the WT post-stroke is a result of memory differences, to revise

Response: As suggested by the reviewer, we revised the sentences in page 20, line 9-10 as: “Moreover, recognition memory assessed with the NOR test also showed significant memory function improvements in these cKO mice”; in page 20, line 20-21: “On the other hand, the locomotor activity in the open field test showed the post-stroke WT animals were hyperactive while the post-stroke cKO mice were not.”

42. Conclusion: The first sentence of the conclusion appears overstated and should be revised

Response: Please see responses to Question 1 of Reviewer #1.

43. Figure 4: The wording related to the phagocytosis of synapses should be revised as mentioned previously

Response: We revised the Fig 4 title in page 39, line 2-3 as: “*Nhe1* cKO brains exhibited increased post-stroke microglial phagocytic activity for dendritic spine plasticity”.

Reviewer #3

1. Cognitive abilities after stroke in WT measuring by Y maze and NOR were not different from those in non-injured mice (Fig 1C and E). According to the authors' previous report published in *Glia*, neurological scores in WT did not recover well until 14 days after MCAO. Can cognitive function be fully restored 28 days after stroke?

Response: Several factors play a role in post-stroke neurological function recovery, such as MCAO duration, infarct volume size, etc. Our 60-min transient MCAO protocol led to ~80 mm³ infarct⁷, with a fully recovery in the Y-maze test by 28 d post-stroke. In literature, mice showing similar infarct after 90-min transient MCAO or permanent MCAO displayed memory deficits in the Y-maze test at 56 d post-stroke⁴⁶. These findings suggest that the faster recovery in our Y-maze study may result from a shorter ischemic period, possibly resulting in reduced level of glutamate, decreased oxidative stress, and less inflammation⁴⁷. On the other hand, our NOR test at 28 d showed a persistent ~68% decrease in the discrimination index in the post-stroke WT, compare to WT post-sham mice, which is in line with previous reports⁴⁸.

2. The number of genes upregulated in cKO is 3349, which is more than 20 times that of WT (Fig2 E). Is there any specific reason? Need an explanation.

Response: To address this question, we conducted new transcriptomic analysis on the stroke-induced differential pathway changes between CL and IL hemispheres from WT or cKO brains using IPA (**Supplementary Figure 9A**). Our results showed that in WT brains, stroke mainly induced upregulation of the coagulation system pathway (**Supplementary Figure 9B**), which was expected in the ischemic stroke condition⁴⁹. In comparison, in the cKO microglia, stroke induced upregulation of several signaling pathways including the eIF2 signaling pathway and oxidative phosphorylation pathway (**Supplementary Figure 9C**). EIF2, an eukaryotic initiation factor, is critical for protein synthesis initiation in translation, and vitally important for antiviral responses⁵⁰. Inactivation of eIF2 was associated with neurodegenerative diseases such as AD⁵¹, and was linked to leukoencephalopathy featured with brain white matter lesion⁵². The mechanisms underlying upregulation of these transcriptions in cKO microglia should be investigated in future studies. The above discussion have been included on page 18, line 21-page 19, line 8.

3. Authors have used anti-P2RY12 to isolate microglia. P2RY12 is one of the homeostatic markers. There are some papers showing P2RY12 mRNA are reduced by microglia activation (such as LPS stimulation). Ref : P2RY12 mRNA is reduced by IFN γ +LPS in microglia from WM and GM (<https://actaneurocomms.biomedcentral.com/articles/10.1186/s40478-019-0850-z>). It would be good to have discussion on it.

Response: Please see responses to Question 1 of Reviewer #2.

4. The balance between Glycolysis and OXPHOS is important for microglial function as authors mentioned in introduction. It is not clear what the status of cKO microglia is because overall glucose metabolism is upregulated. Can upregulated glucose metabolism be beneficial? Are there any references indicating a condition similar to cKO microglia?

Response: Our new Seahorse assays showed that cKO microglia displayed significantly reduced glycolysis and concurrently elevated OXPHOS, which is beneficial for the transformation to restorative microglial functions as reported previously⁵³. Please see responses to Question 1 of Reviewer #1.

5. In Fig.4H, branch points and dendrite length are significantly higher in cKO brain compared to WT even in non-injured brain.

Response: Regarding Reviewer's concerns about increased dendritic plasticity in non-lesion CL hemispheres, we believe that this is due to the elevated microglial phagocytosis. It has been well documented that microglial phagocytosis is important for synapse pruning in healthy mature

brains⁴³⁻⁴⁵. This is supported by our new data in **Figure 4B**, showing that microglial phagocytic activity in the non-stroke CL hemisphere from cKO brains was also significantly increased compared to the WT CL hemisphere, which subsequently impacted synaptic pruning/stripping in non-lesioned brain tissues. We added these new findings in page 10, line 16-20: “In addition, microglial phagocytic activity in the non-stroke CL hemisphere from cKO brains was also significantly increased compared to the WT brains (**Figure 4B**). These *ex vivo* data suggest that selective deletion of microglial *Nhe1* enhances their phagocytic activity in the non-lesion as well as in the acute stroke tissues, which is in line with our RNAseq transcriptomic findings.”

6. Authors showed significantly enhanced myelination of CC in the cKO brains (midline thickness at the same bregma level, Supplementary Figure 4C): How about CC thickness in non-injured brain?

Response: To address the reviewer’s concerns, we have assessed myelin basic protein (MBP) expression in CC of naïve WT and cKO mouse brains. The results show that the CC thickness in the naïve cKO brains are also significantly higher than the naïve WT brains. In addition, the naïve cKO brains showed increased oligodendrogenesis with higher NG2⁺Olig2⁺ OPC counts, and less Caspase3⁺Olig2⁺ apoptotic OLs. These new data suggest that NHE1 activation may play a role in white matter myelination in physiological conditions. These data and description have been included in **Supplementary Figure 7D** and in page 15, line 15-23.

7. Fig4 D. colocalization images are from IL tissues or isolated microglia. Please make a clear in fig legend

Response: We revised the **Figure 4** legend in page 39, line 8-11 : “(C) Representative staining images of C1q, PSD95, and To-pro-3 from the IL peri-lesion area at 3 d post-stroke. Arrows: C1q+/PSD95+ colocalizing cells. (D) Enlarged images of colocalized C1q and PSD95 in the IL peri-lesion brain area (from boxed areas in C).”

8. Fig4H, Add information on the replicates of neurons as well as brains or experiments.

Response: We revised the **Figure 4** legend in page 40, line 5-8 as: “(H) Sholl analysis of branching intersections in relation to distance from soma in neurons in the CL and IL peri-lesion cortex of WT and cKO brains at 28 d post-stroke. N = 24-28 neurons, from 4 brains per group, 3 areas per brain, 2-3 images per area.”

9. In Fig. 4B and 4E showing quantification of microglial phagocytic activity (N=4). What N means? Animals, cells or experiments? Need information on how many cells and animals were used for the graph (for all figure legends).

Response: We revised the **Figure 4** legend in page 39, line 8 as: “(B) Quantification of microglial phagocytic activity. N=4 animals.”; in page 40, line 1: “(E) Quantitative analysis of C1q⁺/PSD95⁺ colocalizing cells. N = 5-6 animals, 3 images per animal.”

Supplementary References

- 1 Persi, E. *et al.* Systems analysis of intracellular pH vulnerabilities for cancer therapy. *Nat Commun* **9**, 2997, doi:10.1038/s41467-018-05261-x (2018).
- 2 Mookerjee, S. A., Goncalves, R. L. S., Gerencser, A. A., Nicholls, D. G. & Brand, M. D. The contributions of respiration and glycolysis to extracellular acid production. *Biochim Biophys Acta* **1847**, 171-181, doi:10.1016/j.bbabi.2014.10.005 (2015).
- 3 Kappler, M., Taubert, H., Schubert, J., Vordermark, D. & Eckert, A. W. The real face of HIF1 α in the tumor process. *Cell Cycle* **11**, 3932-3936, doi:10.4161/cc.21854 (2012).
- 4 Harguindey, S. *et al.* Cariporide and other new and powerful NHE1 inhibitors as potentially selective anticancer drugs--an integral molecular/biochemical/metabolic/clinical approach after one hundred years of cancer research. *J Transl Med* **11**, 282, doi:10.1186/1479-5876-11-282 (2013).
- 5 Rolver, M. G., Elingaard-Larsen, L. O., Andersen, A. P., Counillon, L. & Pedersen, S. F. Pyrazine ring-based Na(+)/H(+) exchanger (NHE) inhibitors potently inhibit cancer cell growth in 3D culture, independent of NHE1. *Sci Rep* **10**, 5800, doi:10.1038/s41598-020-62430-z (2020).
- 6 Previtali, S. C. *et al.* Rimeporide as a first-in-class NHE-1 inhibitor: Results of a phase Ib trial in young patients with Duchenne Muscular Dystrophy. *Pharmacol Res* **159**, 104999, doi:10.1016/j.phrs.2020.104999 (2020).
- 7 Song, S. *et al.* Selective role of Na(+)/H(+) exchanger in Cx3cr1(+) microglial activation, white matter demyelination, and post-stroke function recovery. *Glia* **66**, 2279-2298, doi:10.1002/glia.23456 (2018).
- 8 Valny, M., Honsa, P., Kirdajova, D., Kamenik, Z. & Anderova, M. Tamoxifen in the Mouse Brain: Implications for Fate-Mapping Studies Using the Tamoxifen-Inducible Cre-loxP System. *Front Cell Neurosci* **10**, 243, doi:10.3389/fncel.2016.00243 (2016).
- 9 Fogg, D. K. *et al.* A clonogenic bone marrow progenitor specific for macrophages and dendritic cells. *Science* **311**, 83-87, doi:10.1126/science.1117729 (2006).
- 10 Ajami, B., Bennett, J. L., Krieger, C., Tetzlaff, W. & Rossi, F. M. Local self-renewal can sustain CNS microglia maintenance and function throughout adult life. *Nat Neurosci* **10**, 1538-1543, doi:10.1038/nn2014 (2007).
- 11 Parkhurst, C. N. *et al.* Microglia promote learning-dependent synapse formation through brain-derived neurotrophic factor. *Cell* **155**, 1596-1609, doi:10.1016/j.cell.2013.11.030 (2013).
- 12 Hashimoto, D. *et al.* Tissue-resident macrophages self-maintain locally throughout adult life with minimal contribution from circulating monocytes. *Immunity* **38**, 792-804, doi:10.1016/j.immuni.2013.04.004 (2013).
- 13 Begum, G. *et al.* Selective knockout of astrocytic Na(+)/H(+) exchanger isoform 1 reduces astrogliosis, BBB damage, infarction, and improves neurological function after ischemic stroke. *Glia* **66**, 126-144, doi:10.1002/glia.23232 (2018).
- 14 Lee, E. *et al.* MPTP-driven NLRP3 inflammasome activation in microglia plays a central role in dopaminergic neurodegeneration. *Cell Death Differ* **26**, 213-228, doi:10.1038/s41418-018-0124-5 (2019).

- 15 Zhang, B. *et al.* The specificity and role of microglia in epileptogenesis in mouse models of tuberous sclerosis complex. *Epilepsia* **59**, 1796-1806, doi:10.1111/epi.14526 (2018).
- 16 Schafer, D. P. *et al.* Microglia contribute to circuit defects in *Mecp2* null mice independent of microglia-specific loss of *Mecp2* expression. *Elife* **5**, doi:10.7554/eLife.15224 (2016).
- 17 Ritzel, R. M. *et al.* Functional differences between microglia and monocytes after ischemic stroke. *J Neuroinflammation* **12**, 106, doi:10.1186/s12974-015-0329-1 (2015).
- 18 Liu, Q. & Sorooshyari, S. K. Quantitative and Correlational Analysis of Brain and Spleen Immune Cellular Responses Following Cerebral Ischemia. *Front Immunol* **12**, 617032, doi:10.3389/fimmu.2021.617032 (2021).
- 19 Dotson, A. L. *et al.* Partial MHC Constructs Treat Thromboembolic Ischemic Stroke Characterized by Early Immune Expansion. *Transl Stroke Res* **7**, 70-78, doi:10.1007/s12975-015-0436-4 (2016).
- 20 Alharshawi, K. *et al.* PKC- is dispensable for OX40L-induced TCR-independent Treg proliferation but contributes by enabling IL-2 production from effector T-cells. *Sci Rep* **7**, 6594, doi:10.1038/s41598-017-05254-8 (2017).
- 21 Nemeth, Z. H., Deitch, E. A., Lu, Q., Szabo, C. & Hasko, G. NHE blockade inhibits chemokine production and NF-kappaB activation in immunostimulated endothelial cells. *Am J Physiol Cell Physiol* **283**, C396-403, doi:10.1152/ajpcell.00491.2001 (2002).
- 22 Houlton, J., Zhou, L. Y. Y., Barwick, D., Gowing, E. K. & Clarkson, A. N. Stroke Induces a BDNF-Dependent Improvement in Cognitive Flexibility in Aged Mice. *Neural Plast* **2019**, 1460890, doi:10.1155/2019/1460890 (2019).
- 23 Walker, D. G. *et al.* Patterns of Expression of Purinergic Receptor P2RY12, a Putative Marker for Non-Activated Microglia, in Aged and Alzheimer's Disease Brains. *Int J Mol Sci* **21**, doi:10.3390/ijms21020678 (2020).
- 24 Keren-Shaul, H. *et al.* A Unique Microglia Type Associated with Restricting Development of Alzheimer's Disease. *Cell* **169**, 1276-1290 e1217, doi:10.1016/j.cell.2017.05.018 (2017).
- 25 Krasemann, S. *et al.* The TREM2-APOE Pathway Drives the Transcriptional Phenotype of Dysfunctional Microglia in Neurodegenerative Diseases. *Immunity* **47**, 566-581 e569, doi:10.1016/j.immuni.2017.08.008 (2017).
- 26 Aldana, B. I. Microglia-Specific Metabolic Changes in Neurodegeneration. *J Mol Biol* **431**, 1830-1842, doi:10.1016/j.jmb.2019.03.006 (2019).
- 27 Miron, V. E. Microglia-driven regulation of oligodendrocyte lineage cells, myelination, and remyelination. *J Leukoc Biol* **101**, 1103-1108, doi:10.1189/jlb.3R11116-494R (2017).
- 28 Olah, M. *et al.* Identification of a microglia phenotype supportive of remyelination. *Glia* **60**, 306-321, doi:10.1002/glia.21266 (2012).
- 29 Li, F., Faustino, J., Woo, M. S., Derugin, N. & Vexler, Z. S. Lack of the scavenger receptor CD36 alters microglial phenotypes after neonatal stroke. *J Neurochem* **135**, 445-452, doi:10.1111/jnc.13239 (2015).
- 30 Rinaldi, M. *et al.* Galectin-1 circumvents lysolecithin-induced demyelination through the modulation of microglial polarization/phagocytosis and oligodendroglial differentiation. *Neurobiol Dis* **96**, 127-143, doi:10.1016/j.nbd.2016.09.003 (2016).
- 31 Lampron, A. *et al.* Inefficient clearance of myelin debris by microglia impairs remyelinating processes. *J Exp Med* **212**, 481-495, doi:10.1084/jem.20141656 (2015).

- 32 Miyanishi, K., Sato, A., Kihara, N., Utsunomiya, R. & Tanaka, J. Synaptic elimination by microglia and disturbed higher brain functions. *Neurochem Int* **142**, 104901, doi:10.1016/j.neuint.2020.104901 (2021).
- 33 Wake, H., Moorhouse, A. J., Jinno, S., Kohsaka, S. & Nabekura, J. Resting microglia directly monitor the functional state of synapses in vivo and determine the fate of ischemic terminals. *J Neurosci* **29**, 3974-3980, doi:10.1523/JNEUROSCI.4363-08.2009 (2009).
- 34 Sandvig, I., Augestad, I. L., Haberg, A. K. & Sandvig, A. Neuroplasticity in stroke recovery. The role of microglia in engaging and modifying synapses and networks. *Eur J Neurosci* **47**, 1414-1428, doi:10.1111/ejn.13959 (2018).
- 35 Liu, Y. *et al.* Activation of microglia depends on Na⁺/H⁺ exchange-mediated H⁺ homeostasis. *J Neurosci* **30**, 15210-15220, doi:10.1523/JNEUROSCI.3950-10.2010 (2010).
- 36 Lam, T. I. *et al.* Intracellular pH reduction prevents excitotoxic and ischemic neuronal death by inhibiting NADPH oxidase. *Proc Natl Acad Sci U S A* **110**, E4362-4368, doi:10.1073/pnas.1313029110 (2013).
- 37 Courtney, R. & Landreth, G. E. LXR Regulation of Brain Cholesterol: From Development to Disease. *Trends Endocrinol Metab* **27**, 404-414, doi:10.1016/j.tem.2016.03.018 (2016).
- 38 Hammond, T. R. *et al.* Single-Cell RNA Sequencing of Microglia throughout the Mouse Lifespan and in the Injured Brain Reveals Complex Cell-State Changes. *Immunity* **50**, 253-271 e256, doi:10.1016/j.immuni.2018.11.004 (2019).
- 39 Bialas, A. R. & Stevens, B. TGF-beta signaling regulates neuronal C1q expression and developmental synaptic refinement. *Nat Neurosci* **16**, 1773-1782, doi:10.1038/nn.3560 (2013).
- 40 Hong, S. *et al.* Complement and microglia mediate early synapse loss in Alzheimer mouse models. *Science* **352**, 712-716, doi:10.1126/science.aad8373 (2016).
- 41 Guzman, K. M. *et al.* Conditional depletion of Fus in oligodendrocytes leads to motor hyperactivity and increased myelin deposition associated with Akt and cholesterol activation. *Glia* **68**, 2040-2056, doi:10.1002/glia.23825 (2020).
- 42 Kettenmann, H., Kirchhoff, F. & Verkhratsky, A. Microglia: new roles for the synaptic stripper. *Neuron* **77**, 10-18, doi:10.1016/j.neuron.2012.12.023 (2013).
- 43 Schafer, D. P. *et al.* Microglia sculpt postnatal neural circuits in an activity and complement-dependent manner. *Neuron* **74**, 691-705, doi:10.1016/j.neuron.2012.03.026 (2012).
- 44 Gyorffy, B. A. *et al.* Local apoptotic-like mechanisms underlie complement-mediated synaptic pruning. *Proc Natl Acad Sci U S A* **115**, 6303-6308, doi:10.1073/pnas.1722613115 (2018).
- 45 Stevens, B. *et al.* The classical complement cascade mediates CNS synapse elimination. *Cell* **131**, 1164-1178, doi:10.1016/j.cell.2007.10.036 (2007).
- 46 Tanaka, Y. *et al.* Early Reperfusion Following Ischemic Stroke Provides Beneficial Effects, Even After Lethal Ischemia with Mature Neural Cell Death. *Cells* **9**, doi:10.3390/cells9061374 (2020).
- 47 Belov Kirdajova, D., Kriska, J., Tureckova, J. & Anderova, M. Ischemia-Triggered Glutamate Excitotoxicity From the Perspective of Glial Cells. *Front Cell Neurosci* **14**, 51, doi:10.3389/fncel.2020.00051 (2020).

- 48 Pan, X. *et al.* Physical Exercise Promotes Novel Object Recognition Memory in Spontaneously Hypertensive Rats after Ischemic Stroke by Promoting Neural Plasticity in the Entorhinal Cortex. *Front Behav Neurosci* **11**, 185, doi:10.3389/fnbeh.2017.00185 (2017).
- 49 van Os, H. J. A. *et al.* Intrinsic Coagulation Pathway, History of Headache, and Risk of Ischemic Stroke. *Stroke* **50**, 2181-2186, doi:10.1161/STROKEAHA.118.023124 (2019).
- 50 Shrestha, N. *et al.* Eukaryotic initiation factor 2 (eIF2) signaling regulates proinflammatory cytokine expression and bacterial invasion. *J Biol Chem* **287**, 28738-28744, doi:10.1074/jbc.M112.375915 (2012).
- 51 Oliveira, M. M. *et al.* Correction of eIF2-dependent defects in brain protein synthesis, synaptic plasticity, and memory in mouse models of Alzheimer's disease. *Sci Signal* **14**, doi:10.1126/scisignal.abc5429 (2021).
- 52 Li, W., Wang, X., Van Der Knaap, M. S. & Proud, C. G. Mutations linked to leukoencephalopathy with vanishing white matter impair the function of the eukaryotic initiation factor 2B complex in diverse ways. *Mol Cell Biol* **24**, 3295-3306, doi:10.1128/MCB.24.8.3295-3306.2004 (2004).
- 53 Lauro, C. & Limatola, C. Metabolic Reprograming of Microglia in the Regulation of the Innate Inflammatory Response. *Front Immunol* **11**, 493, doi:10.3389/fimmu.2020.00493 (2020).

Reviewers' comments:

Reviewer #1 (Remarks to the Author):

The authors have adequately addressed points 3 - 10 of the initial review, but not responses to points 1 and 2 are incomplete

Original point #1 pertains to the authors' claims of cause effect vs. correlations among the outcome measures reported. The use of the word "for" in the title, abstract, and elsewhere makes it ambiguous as to what the authors are concluding. For example, in the title it is not still not clear whether the authors are stating that the increase in microglial oxidative phosphorylation is the reason that these animals show increased post-stroke remyelination and cognitive function recovery. This is implied, but the data presented do not exclude other potential effects of NHE1 deletion on these changes. It is not obvious that a direct causal relationship does exist, as there is no evidence that microglial ATP production is the rate-limiting step for phagocytic activity, and in fact the gene upregulation data suggests that it is not. Please clarify here and the other places where the word "for" is used (abstract, figure legends, etc.) where a cause-effect relationship is being stated.

Original point #2 pertains to a crucial methodological detail, i.e. proximity of the areas analyzed to the infarct margin. As noted in the original review, this is a crucial point because there is a gradient of effects on neuronal synapse loss and other markers from the lesion edge outwards, and so observations will depend on proximity of the photographed region to the infarct margin. The diagram added to panel C of figure 4 does not help in this regard, as it shows a range of areas extending from and including the infarcted region (which presumably was not actually sampled). What is needed is a description in the methods that clarifies how the lesion edge was defined and identified, and what exactly was the spatial relationship between the lesion edge and region(s) analyzed. It may be that the experiments were not done this way, but instead the same general region was photographed in each brain by observers blinded to the mouse genotype. In that case, the methods should state the coordinates or landmarks that the photographer used to decide where to photograph.

Reviewer #2 (Remarks to the Author):

The authors have addressed all of our concerns. The manuscript has improved substantially.

We only have two very minor comments at this stage:

Page 8 line 7

I would avoid the term polarized as it is a direct reference to the M1/M2 nomenclature which was already changed in the manuscript: Suggestion: "represent the microglia population in WT and cKO in control and stroke contexts."

These findings collectively indicate that our isolated P2RY12 7 + microglial population from stroke brains represent homeostatic as well as polarized microglia

Page 12 line 2

Importantly, a dramatic reduction in spine density was observed in the WT mice compared to the 3 cKO mice ($p < 0.0001$; Figure 4K-L),

I would just clarify that you are talking about the post-recovery point because it might be confusing.

Reviewer #3 (Remarks to the Author):

Now the manuscript is acceptable for publication.

We would like to thank our reviewers for their critical reviews. We have thoroughly revised the manuscript according to the reviewers' suggestions. Please see below for our point-by-point responses.

Reviewer #1

The authors have adequately addressed points 3 - 10 of the initial review, but not responses to points 1 and 2 are incomplete

Original point #1 pertains to the authors' claims of cause effect vs. correlations among the outcome measures reported. The use of the word "for" in the title, abstract, and elsewhere makes it ambiguous as to what the authors are concluding. For example, in the title it is not still not clear whether the authors are stating that the increase in microglial oxidative phosphorylation is the reason that these animals show increased post-stroke remyelination and cognitive function recovery. This is implied, but the data presented do not exclude other potential effects of NHE1 deletion on these changes. It is not obvious that a direct causal relationship does exist, as there is no evidence that microglial ATP production is the rate-limiting step for phagocytic activity, and in fact the gene upregulation data suggests that it is not. Please clarify here and the other places where the word "for" is used (abstract, figure legends, etc.) where a cause-effect relationship is being stated.

Response: We agree with the reviewer's concerns as we are unable to conduct a causal relationship analysis with our current experimental designs (according to Altman et. Al. ¹). As suggested by the reviewer, we revised the title to: "Stimulating microglial oxidative phosphorylation and phagocytosis in post-stroke brain repair and cognitive function recovery". Therefore, we also corrected wording indicating such causal relationships throughout the whole text, as listed below:

- page 3, line 12-15: "This study reveals that genetic blockade of microglial NHE1 stimulated OXPHOS immunometabolism, and boosted phagocytosis function which is associated with tissue remodeling and post-stroke cognitive function recovery";
- page 5, line 23-line 6, line 1: "These novel findings suggest that microglial OXPHOS metabolism and elevated phagocytosis functions likely collectively enhances brain tissue remodeling and post-stroke cognitive function recovery";
- page 10, line 9-10: "Microglial *Nhe1* cKO brains exhibited increased post-stroke microglial phagocytic activity and improved dendritic spine plasticity";

- page 12, line 10-12: “These findings provide additional evidence that elevated microglial phagocytosis in the cKO brains at the early phase post-stroke could facilitate synaptic stripping for better synaptic remodeling”;
- page 14, line 9-12: “These data strongly suggest that loss of microglial NHE1 protein in the *Nhe1* cKO brains plays an important role in fine-tuning microglial glycolytic and OXPHOS metabolism to provide ATP fuels in meeting their energy demands in phagocytic functions”;
- page 21, line 2-5: “Our findings strongly suggest that selective deletion of microglial *Nhe1* promoted stripping of dendritic spines and synapses through increased microglial phagocytosis, which could contribute to faster post-stroke cognitive function recovery”;
- page 21, line 16-19: “We suspect that this was largely resulted from early microglia-mediated phagocytosis of myelin debris in the cKO brains. Additional studies are needed to determine the temporal course of the microglial phagocytic activity and correlation to white matter repair”;
- page 40, line 2-3: “*Nhe1* cKO brains exhibited increased post-stroke microglial phagocytic activity and improved dendritic spine plasticity”;
- page 46, line 2-3: “Illustration of NHE-1-mediated microglial pH_i regulation and immunometabolism in ischemic brain repair”;
- page 46, line 5-7: “Selective deletion of microglial *Nhe1* acidifies pH_i , which is involved in boosting the immunometabolism of OXPHOS, as well as ATP production, which provides fuels for important microglial functions such as phagocytosis.”;
- page 46, line 12-13: “These changes may collectively contribute to cognitive function improvement after brain lesions”.

Original point #2 pertains to a crucial methodological detail, i.e. proximity of the areas analyzed to the infarct margin. As noted in the original review, this is a crucial point because there is a gradient of effects on neuronal synapse loss and other markers from the lesion edge outwards, and so observations will depend on proximity of the photographed region to the infarct margin. The diagram added to panel C of figure 4 does not help in this regard, as it shows a range of areas extending from and including the infarcted region (which presumably was not actually sampled). What is needed is a description in the methods that clarifies how the lesion edge was defined and identified, and what exactly was the spatial relationship between the lesion edge and region(s) analyzed. It may be that the experiments were not done this way, but instead the same general region was photographed in each brain by observers blinded to the mouse genotype. In

that case, the methods should state the coordinates or landmarks that the photographer used to decide where to photograph.

Response: As suggested by the reviewer, we have included detailed methods for image collection in **Supplementary Materials**, page 9, line 17-20: “Under 40× objective lens using an Olympus IX81 confocal microscope (Olympus, Japan), fluorescent images were captured in perilesion areas which were defined as 150-450 μm from the margin of ischemic core showing condensed nuclei or tissue losses.”

Reviewer #2

The authors have addressed all of our concerns. The manuscript has improved substantially.

We only have two very minor comments at this stage:

Page 8 line 7

I would avoid the term polarized as it is a direct reference to the M1/M2 nomenclature which was already changed in the manuscript: Suggestion: “represent the microglia population in WT and cKO in control and stroke contexts.”

These findings collectively indicate that our isolated P2RY12 + microglial population from stroke brains represent homeostatic as well as polarized microglia

Response: We thank the reviewer for the suggestion. We have revised the sentence in **page 8, line 7-9**: “These findings collectively indicate that our isolated P2RY12⁺ microglial population from WT and cKO stroke brains represent the microglia population in both control and stroke contexts.”

Page 12 line 2

Importantly, a dramatic reduction in spine density was observed in the WT mice compared to the cKO mice (p < 0.0001; Figure 4K-L),

I would just clarify that you are talking about the post-recovery point because it might be confusing.

Response: We have revised the sentence in **page 12, line 4-5**: “Importantly, a dramatic reduction in spine density was observed in the WT mice compared to the cKO mice at 28 d post-stroke (p < 0.0001; Figure 4K-L)...”

Reviewer #3

Now the manuscript is acceptable for publication.

References

- 1 Altman, N. & Krzywinski, M. Association, correlation and causation. *Nat Methods* **12**, 899-900, doi:10.1038/nmeth.3587 (2015).